



Atmospheric
Chemistry
and Physics

# Effects of black carbon mitigation on Arctic climate

**Thomas Kühn**[1,2]**, Kaarle Kupiainen**[3,a]**, Tuuli Miinalainen**[1]**, Harri Kokkola**[2]**, Ville-Veikko Paunu**[3]**, Anton Laakso**[2]**,**
**Juha Tonttila**[2]**, Rita Van Dingenen**[4]**, Kati Kulovesi**[5]**, Niko Karvosenoja**[3]**, and Kari E. J. Lehtinen**[1,2]

[1]Department of Applied Physics, University of Eastern Finland (UEF), Kuopio, Finland
[2]Atmospheric Research Centre of Eastern Finland, Finnish Meteorological Institute (FMI), Kuopio, Finland
[3]Finnish Environment Institute (SYKE), Helsinki, Finland
[4]European Commission, Joint Research Centre (JRC), Ispra (VA), Italy
[5]School of Law, University of Eastern Finland (UEF), Joensuu, Finland
[a]currently at: the Finnish Ministry of the Environment, Helsinki, Finland

**Correspondence:** Thomas Kühn (thomas.h.kuhn@uef.fi)

**Abstract.** We use the ECHAM-HAMMOZ aerosol-climate model to assess the effects of black carbon (BC) mitigation measures on Arctic climate. To this end we constructed several mitigation scenarios that implement all currently existing legislation and then implement further reductions of BC in a successively increasing global area, starting from the eight member states of the Arctic Council, expanding to its active observer states, then to all observer states, and finally to the entire globe. These scenarios also account for the reduction of the co-emitted organic carbon (OC) and sulfate (SU). We find that, even though the additional BC emission reductions in the member states of the Arctic Council are small, the resulting reductions in Arctic BC mass burdens can be substantial, especially in the lower troposphere close to the surface. This in turn means that reducing BC emissions only in the Arctic Council member states can reduce BC deposition in the Arctic by about 30 % compared to the current legislation, which is about 60 % of what could be achieved if emissions were reduced globally. Emission reductions further south affect Arctic BC concentrations at higher altitudes and thus only have small additional effects on BC deposition in the Arctic. The direct radiative forcing scales fairly well with the total amount of BC emission reduction, independent of the location of the emission source, with a maximum direct radiative forcing in the Arctic of about $-0.4\,\mathrm{W\,m^{-2}}$ for a global BC emission reduction. On the other hand, the Arctic effective radiative forcing due to the BC emission reductions, which accounts for aerosol–cloud interactions, is small compared to the direct aerosol radiative forcing. This happens be-

cause BC- and OC-containing particles can act as cloud condensation nuclei, which affects cloud reflectivity and lifetime and counteracts the direct radiative forcing of BC. Additionally, the effective radiative forcing is accompanied by very large uncertainties that originate from the strong natural variability of meteorology, cloud cover, and surface albedo in the Arctic. We further used the TM5-FASST model to assess the benefits of the aerosol emission reductions for human health. We found that a full implementation in all Arctic Council member and observer states could reduce the annual global number of premature deaths by 329 000 by the year 2030, which amounts to 9 % of the total global premature deaths due to particulate matter.

## 1 Introduction

Black carbon (BC) is emitted into the atmosphere as microscopically small, solid particles formed as a result of incomplete combustion (Goldberg, 1985). The climate effects of atmospheric BC are complex. As an efficient light-absorbing compound it is generally thought to warm the climate (Ramanathan and Carmichael, 2008). This effect becomes very important in the Arctic, because atmospheric light absorption is enhanced above the reflecting snow and ice surfaces and also because the deposited BC particles darken the snow and ice, which affects the melt rate (AMAP, 2015). On the other hand, the ageing of BC aerosol particles increases their hygroscopicity and makes them potential cloud condensation

nuclei (CCN) (Kuwata et al., 2009). While increases in CCN will have a cooling effect, BC in cloud droplets or nearby clouds tends to warm the cloud, affecting the evaporation of clouds as well as the atmospheric stability, which leads to changes in cloud dynamics. This results in a semi-direct cooling effect (Koch and Del Genio, 2010). Eventually, as BC is deposited on snow and ice, it increases the melt rate and contributes to the thinning of glaciers and loss of Arctic sea ice (Menon et al., 2010; AMAP, 2015). In addition, changes in BC emissions usually also affect the emission of other, co-emitted aerosol compounds, like organic carbon and sulfate (Klimont et al., 2017). These species mostly scatter light, thereby reflecting part of the incoming sunlight, which leads to cooling (Kiehl and Briegleb, 1993). They also, like aged BC, can act as CCN and thus affect cloud properties (Twomey, 1977; Albrecht, 1989). Altogether this makes it very hard to assess the climatic effects of BC mitigation.

Apart from climate effects, aerosol mitigation is very important for enhancing air quality in many regions of the world, which affects many aspects of life, the most important of which is health. As humans (and animals) inhale aerosol (usually the measure is particulate matter with diameters below $2.5\,\mu\text{m}$ or $PM_{2.5}$) for long periods of time, part of the aerosol mass deposits in the respiratory tract and may even enter the bloodstream (Kim et al., 2015). This can severely increase the risk of developing many kinds of diseases, including respiratory diseases, heart diseases, and strokes (Pope III et al., 2002; Krewski et al., 2009; Anenberg et al., 2012). When assessing the importance of BC mitigation, the co-benefits of these aspects should hence be taken into account (Partanen et al., 2018).

Emissions from within the Arctic area (which we here define as 60–90° north) account for only a small fraction of the global emissions, and most of the impacts are induced by BC emitted and imported from outside the area (Winiger et al., 2019). Recent studies have indeed indicated that an important pathway of BC contributing to Arctic warming is through the transport of heated air masses from outside the area, especially from mid-latitudes (Yang et al., 2014; Sand et al., 2016).

Different emission sectors contribute differently to the total BC emissions in different parts of the world. Globally, burning of fossil fuels and biomass in transport, household heating, and cooking as well as wildfires are important emission sources of BC. In the Arctic Council member states, on the other hand, the key anthropogenic emission sources include transport and household heating as well as flaring in the oil and gas industry (AMAP, 2015, 2019). Arctic shipping is currently a relatively minor source, but its relative importance is projected to increase with the decrease in the Arctic sea ice extent (Stohl et al., 2013).

The recent AMAP assessment (AMAP, 2015) indicated that with targeted choices of already existing mitigation measures of BC-rich sources, it could be possible to cut the projected global and Arctic climate impacts significantly in the coming few decades, provided that they could be implemented globally on a large scale. Such reductions can, however, be politically very demanding to achieve, since currently no mechanisms or policy processes are in place. At the international level, there are no legally binding mitigation measures applicable to BC, apart from commitments to reduce BC as part of fine particulate matter ($PM_{2.5}$) under the Gothenburg Protocol to the Convention on Long-Range Transboundary Air Pollution (Gothenburg Protocol, 1999). However, important non-binding processes to accelerate regional action exist under the Arctic Council. The key examples include the framework document "Enhanced Black Carbon and Methane Emissions Reductions, An Arctic Council Framework for Action", adopted by the Arctic Council in their 2015 meeting (Arctic Council, 2015). According to the document, the Arctic Council member states are committed to accelerating the decline in BC emissions and call upon the Arctic Council observer states to participate in the efforts. Currently eight observer states have participated in the process. Furthermore, the 10th Arctic Council Ministerial Meeting in May 2017 adopted an aspirational collective goal of limiting BC emissions between 25 % and 33 % below 2013 levels by 2025 (Arctic Council, 2017). In addition to these non-binding formal frameworks, voluntary action can also be driven by co-benefits at the local scale, which include air quality, human health, and crop yields.

In this work we study what could be achieved by accelerated BC actions in the Arctic Council member states alone and together with the observer states in terms of reducing atmospheric burden, deposition, and radiative forcing of BC in the Arctic. The analysis takes into account the cooling by co-emitted sulfur species and organic carbon. The results are compared with large-scale global emission reduction scenarios that have been the foundation of previous studies. The study brings to light the unique and still relatively unexplored institutional potential of the Arctic Council to catalyse global regulatory action on the abatement of air pollution by engaging its observer states in concrete, quantitative, and collective actions on BC reduction.

## 2 Methods

### 2.1 Emission scenarios

As anthropogenic emission inputs we used the ECLIPSE version 5a emission scenarios, which include data for black carbon (BC), organic carbon (OC) or organic matter (OM), sulfur dioxide ($SO_2$), methane ($CH_4$), carbon monoxide (CO), nitrogen oxides ($NO_x$), non-methane volatile organic compounds (nmVOCs), and ammonia ($NH_3$) from the IIASA-GAINS model (Stohl et al., 2015; Klimont et al., 2017). The emissions are available in 5-year intervals, spatially distributed onto a $0.5° \times 0.5°$ latitude grid, and include monthly data for the major sectors. For the present study, we only used

emission data for the major aerosol compounds BC, OC, and $SO_2$ and re-gridded the data to the T63 model resolution, which roughly corresponds to $2° \times 2°$.

We utilized particularly two of the scenarios, namely the Current Legislation (BASELINE) scenario and the short-lived climate forcer (SLCF) mitigation (MITIGATE) scenario, as starting points to construct the emission data sets for this study. The BASELINE scenario assumes that all 2015 agreed legislation and adopted policies affecting air pollutant emissions (see e.g. Cofala et al., 2007, and AMAP, 2015) will be implemented. The SLCF mitigation scenario (MITIGATE) additionally assumes the full global implementation of SLCF emission reduction technologies phasing in by 2030 and 2050 (see Shindell et al., 2012). The technologies were selected from existing emission control options for particulate and gaseous species in the GAINS model by assessing the potential climate impact using a climate metric (Shindell et al., 2012; Stohl et al., 2015) and can therefore be viewed as a maximum feasible SLCF reduction scenario.

For the purposes of this study we constructed combinations of the BASELINE and MITIGATE scenarios to study the impact of emission reduction measures taken by the member states of the Arctic Council member and observer states (see Table 1 and Fig. 1) on the Arctic climate. As mentioned above, AMAP (2015) and Stohl et al. (2015) have used similar data sets, but they introduced the emission reductions globally, whereas in this work we apply the emission reductions in successively larger regions of the globe. As a reference scenario we used the ECLIPSE BASELINE scenario (here referred to as CLE). We further constructed scenarios where the additional MITIGATE SLCF reductions are implemented:

1. in the Arctic Council member states (AC),

2. in the Arctic Council member and active observer states (AC_ACT; countries that have shown interest in joining the Framework for Action on Black Carbon and Methane by submitting a national report to the Arctic Council in 2015),

3. in the Arctic Council member states and all observer states (AC_ALL), and

4. globally (GLOB; equal to the ECLIPSE MITIGATION scenario).

The global extents of the implemented SLCF emission reductions for the different scenarios are outlined in Fig. 1. Ship emissions are included in the ECLIPSE scenarios, but it is unclear how individual countries can affect these emissions, and they are therefore the same in all scenarios. All emission data sets not covered by the ECLIPSE emissions (i.e. aircraft emissions, biogenic emissions, and wildfires) were taken from the ECHAM-HAMMOZ standard emission data sets (Granier et al., 2011; Diehl et al., 2012).

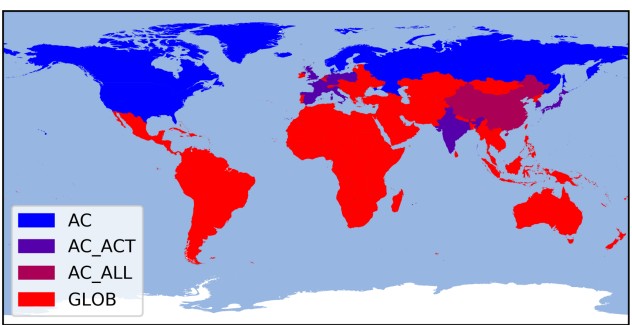

**Figure 1.** Global extent of SLCF reductions for the different scenarios. Starting from AC, each scenario includes all countries of the previous scenario (e.g. AC_ACT = AC plus active observer states).

## 2.2 Aerosol-climate model

For our climate simulations, we used the ECHAM-HAMMOZ aerosol-climate model (ECHAM6.3-HAM2.3-MOZ1.0). The ECHAM6 host atmospheric model (Stevens et al., 2013) computes the atmospheric circulation and fluxes using a semi-Lagrangian transport scheme, HAM (Tegen et al., 2019) models aerosol processes, and MOZ (Schultz et al., 2018) (not used in this study) models atmospheric chemistry. Aerosol emissions, transport, radiation interaction, and water update are modelled with HAM. Within HAM, two different aerosol microphysics models can be used: either the M7 modal aerosol module (Tegen et al., 2019) or the Sectional Aerosol module for Large Scale Applications (SALSA) (Kokkola et al., 2018). Here we use SALSA to solve the aerosol microphysics (hereafter we refer to this model setup as ECHAM-SALSA); SALSA represents aerosols by dividing the aerosol size distribution into 10 size sections (or bins), where the aerosol population is further divided into a soluble and an insoluble sub-population. A detailed description of the SALSA size distribution is given in Kokkola et al. (2018), elaborating on the size resolution and which aerosol compounds are treated in which size bin. In the same article, an evaluation of ECHAM-SALSA against satellite and ground-based remote sensing instruments, in situ observations of aerosol composition and size distribution, as well as aircraft measurements of aerosol composition has been performed. In addition, ECHAM-SALSA has been involved in several model experiments within the AEROCOM initiative, where models are compared against aerosol observations and against each other (e.g. Burgos et al., 2020; Kristiansen et al., 2016; Kipling et al., 2016; Tsigaridis et al., 2014). Furthermore, ECHAM-SALSA's capability to simulate aerosol–cloud interactions compared to satellite observations has been evaluated in a previous study by Saponaro et al. (2020). SALSA treats the chemical species sulfate (SU), organic carbon (OC), black carbon (BC), sea salt (SS), and mineral dust (DU). Within one size bin of one sub-population, all aerosol particles are assumed to have the

**Table 1.** The Arctic Council member and observer states (before 2017).

| Arctic Council members | Canada, Denmark, Finland, Iceland, Norway, Russia, Sweden, USA |
|---|---|
| Active observers | France, Germany, India, Italy, Japan, Poland, South Korea, Spain, United Kingdom |
| Other observers | China, the Netherlands, Singapore |

same chemical composition, while the two sub-populations are treated as externally mixed. SALSA solves the aerosol processes of nucleation, condensation, coagulation, activation into cloud droplets, and aerosol removal. HAM includes a simplified sulfate chemistry, which oxidises gaseous sulfur dioxide ($SO_2$) into sulfuric acid ($H_2SO_4$), which can either nucleate to form new particles or condense onto existing particles; 2.5 % of the total $SO_2$ emissions are converted into $SO_4$ at emission time and released as primary particles. In SALSA cloud droplet activation is solved using the parameterisation by Abdul-Razzak and Ghan (2002) such that both soluble and insoluble particles can form cloud droplets. In the cloud activation routine, SU and OC are treated as fully dissolved compounds, with hygroscopicity values ($\kappa$) of 0.57 and 0.21, respectively. BC is assumed to be completely insoluble and contributes to cloud droplet activation only indirectly, by facilitating condensation of sulfuric acid to the particle phase.

## 2.3  Climate simulations

The scenarios that were used in this study are outlined in Sect. 2.1. For each scenario, we considered the year 2010 to be the present; thus, emission strengths are the same for all scenarios for 2010 and diverge after that. All SLCF emission reductions are assumed to be fully implemented by 2050, and we thus performed two simulations per scenario, one for the year 2030 and one for the year 2050. Together with the reference simulation for 2010, this makes a total of 11 simulations. In order to ensure sufficient statistics, each simulation was run for 30 years plus half a year of spin-up.

As we here were only interested in assessing the aerosol effect on the Arctic climate, the atmospheric greenhouse gas mixing ratios were set to the values of the year 2010 for all simulations. The values used were based on the Representative Concentration Pathway 4.5 scenario (RCP4.5), following the fifth assessment report of the International Panel for Climate Change (IPCC) (IPCC, 2013) (note that for the year 2010 the greenhouse gas concentrations are almost identical for all four RCPs, and therefore the choice of any particular RCP has no influence on the findings in this study). This means that the mixing ratios for $CO_2$, $CH_4$, and $N_2O$ were set to 389.1 ppm, 1767 ppb, and 323 ppb, respectively. Furthermore, the sea surface temperatures (SST), sea ice cover (SIC), and (spatially varying) ozone concentrations were the same in all simulations. For SST and SIC we used the monthly varying climatologies from the PCMDI's Atmo-

spheric Model Intercomparison Project (Taylor et al., 2012) for the year 2010. For the 3D ozone and OH concentrations we used the reanalysis of the atmospheric oxidants for the year 2010 as described in Inness et al. (2013). It should be mentioned that fixing greenhouse gas concentrations, SST, and SIC in this way prohibits several feedback mechanisms that may affect the magnitude of the simulated aerosol radiative forcing, mainly because atmospheric temperatures are not allowed to adjust freely. These effects include for instance changes in atmospheric water vapour content, which may affect clouds and thereby the atmospheric aerosol concentrations, and non-anthropogenic (e.g. biogenic) aerosol emissions, which may also affect the anthropogenic aerosol in several ways.

In all simulations performed in this study, the horizontal model resolution was set to the T63 spectral truncation, which corresponds to a resolution of roughly $2° \times 2°$, and a vertical resolution of 47 hybrid sigma-pressure levels was used. The model meteorology was allowed to evolve freely. This together with the fixed SST and SIC allows for rapid adjustments of the atmosphere while avoiding climate feedbacks and therefore makes it possible to calculate the effective radiative forcing (ERF) (Lohmann et al., 2010; Forster et al., 2016). The ERF is calculated as the difference of the average net radiative flux at the top of the atmosphere between the reference simulation (2010) and any of the simulations using emissions from a reduction scenario. The aerosol direct radiative effect (DRE) is calculated online by performing the radiation calculations twice, once with and once without accounting for aerosol–radiation interaction. The aerosol direct radiative forcing (DRF) is then again computed as the difference in DRE between the reference simulation (2010) and any of the simulations using emissions from a reduction scenario.

## 2.4  Human health and mortality evaluation

We utilized the Tracer Model 5 Fast Scenario Screening Tool (TM5-FASST), developed at JRC Ispra (Italy), to assess the impact of the different mitigation scenarios outlined in Sect. 2.1 on human health. TM5-FASST evaluates how air pollutant emissions affect large-scale pollutant concentrations and their impact on human health (e.g. mortality and years of life lost) and crop yield. It utilizes source–receptor relationships to link emissions of pollutants in a given source region to downwind concentrations and related impacts. The source–receptor relationships have been derived by utilizing

a large amount of simulations with the TM5 chemical transport model (Huijnen et al., 2010), which accounts for the effects of meteorology and chemical and physical processes on the transport of particulate matter (PM) (Van Dingenen et al., 2018). The source–receptor relationships were derived using present-day meteorological data and were fixed for all the scenarios investigated here. Changes in aerosol concentrations can affect meteorology, which can feed back on certain aerosol processes, most notably wet removal, and thereby transport of aerosol. By fixing the meteorology, these effects are effectively ignored. However, as aerosols have comparably short atmospheric lifetimes, aerosol sources affect PM surface concentrations close-by the most, and the resulting error should therefore be relatively small. Health impacts from particulate matter with diameter smaller than $2.5\,\mu m$ ($PM_{2.5}$) are calculated as the number of annual premature mortalities from five causes of death, following the Global Burden of Diseases (GBD) methodology (Lim et al., 2012): ischemic heart disease, chronic obstructive pulmonary disease, stroke, lung cancer, and acute lower respiratory airway infections. More details on the model can be found in Van Dingenen et al. (2018).

## 2.5 Uncertainty intervals and statistical significance

Unless stated otherwise, the values provided in the text and figures are arithmetical average values over all analysed simulation years and uncertainty intervals are presented as 1 standard deviation. For some of the results we additionally performed a two-sided Mann–Whitney $U$ test to analyse statistical significance. In these cases we assume that a $p$ value of less than 0.05 denotes a statistically significant difference between the results of two simulations.

## 3 Results

### 3.1 Global emissions

Figure 2 shows the change in the global anthropogenic emissions of black carbon (BC), organic carbon (OC), and sulfate ($SU = SO_2 + SO_4$), which are the anthropogenic aerosol species that are modelled by ECHAM-SALSA. The first thing to note is the way that the different mitigation scenarios affect the aerosol emissions. The black line in each plot shows the effect of the current legislation (CLE) from years 2010 to 2050. Here SU shows the strongest changes in emission strength (about 19 % reduction by 2030), while OC and BC emissions change less (about 14 % and 13 % reduction by 2030, respectively). The SLCF mitigation scenarios include additional emission reduction measures to the CLE scenario for 2030 and 2050. As SU has an overall cooling effect in the atmosphere, it is usually considered unfavourable to reduce SU emissions when trying to slow global warming. Therefore, the SLCF mitigation measures have been selected such that they are mainly SU-neutral. In Fig. 2 this can

be seen by comparing the different scenarios for the same simulation year: while SU emissions show very little further change from the CLE scenario (less than 0.5 %; note how the lines in Fig. 2c lie on top of each other), BC emissions decrease dramatically, with a maximum reduction since 2010 of 81.3 % in 2050 for global implementation of the reductions (scenario GLOB). This amounts to decreasing the global anthropogenic BC emissions by 78.8 % in 2050 when comparing scenario GLOB to scenario CLE. As OC and BC are usually co-emitted species, the OC emissions decrease accordingly (70.7 % from 2010 to 2050 for scenario GLOB). Note in Fig. 2c how the anthropogenic emission strength of SU almost recovers to the value of 2010 between 2030 and 2050, which can be attributed to economic growth. Here it should be noted that, while the global total emissions increase, regional trends, especially in the developed world (e.g. in Europe and North America), may be of opposite sign. In comparison, the changes in BC and OC emissions between 2030 and 2050 for the different emission scenarios are much smaller (and not necessarily of the same direction). This occurs because in the SLCF mitigation scenarios the mitigation measures are assumed to be fully implemented by 2030 and, furthermore, the largest parts of BC and OC emissions come from different sectors than SU, which develop differently with time.

### 3.2 Arctic aerosol burdens

Depending on the emission site, pollutants can reach the Arctic through different transport pathways. While pollutants emitted within the Arctic are mostly transported close to the surface, pollutants from sources further south mostly enter the Arctic at relatively high altitudes (Sobhani et al., 2018; Winiger et al., 2019). Uplifting of these pollutants happens either directly after emission or when the pollutants reach the polar dome. Within the polar dome, vertical exchange of air masses is very slow (Stohl, 2006; Quinn et al., 2011). In our simulations, this difference in transport pathways is clearly visible when analysing the vertical aerosol profiles over the Arctic region (60–90° north). Figure 3a–c show the Arctic yearly average vertical aerosol mass distribution profile for BC, OC, and SU, respectively. All three profiles show two peaks as a function of altitude – one close to the surface and one at approximately 200 hPa, which is above the Arctic tropopause. Each profile also shows a pronounced minimum between 400 and 500 hPa, depending on the species. As the vertical location of aerosols within the troposphere is important for many atmospheric processes (e.g. aerosol–cloud interactions and aerosol deposition), we divided the Arctic atmosphere into a lower troposphere (LT) and a rest of the atmosphere (RA) part, using the minimum of the BC profile (at approximately 450 hPa) as a limit between the two. In Kokkola et al. (2018), the BC, OC, and SU vertical profiles modelled by ECHAM-SALSA were compared to several aircraft campaigns. There it was found that ECHAM-SALSA

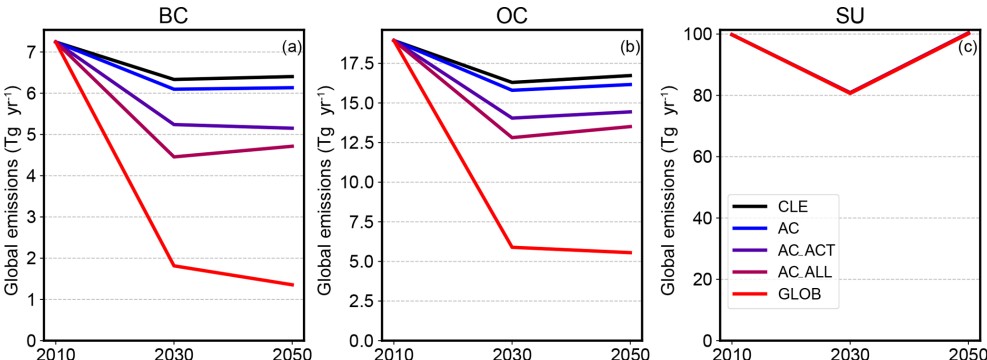

**Figure 2.** Total yearly global anthropogenic emission of BC **(a)**, OC **(b)**, and SU **(c)** for the different emission scenarios. The colouring for the scenarios is the same as in Fig. 1, with CLE plotted in black.

tends to overestimate BC concentrations in the source regions and underestimate BC concentrations at high latitudes. Furthermore, we compared the modelled BC vertical profiles to measurement data from the ATom and HIPPO campaigns (not shown), where the model compares quite well with the observations at all latitudes. The OC and SU modelled concentrations agreed in most cases much better with the observations.

Figure 3d–f show how the vertical profiles change from 2010 to 2030 for the different scenarios, while Fig. 3g–i show the changes from 2010 to 2050. As explained above, most of the aerosols emitted within the Arctic contribute to the LT concentrations. Accordingly, when reducing emissions in the Arctic Council member states (scenario AC), BC and OC concentrations decrease the most close to the surface, while the RA concentration changes are much smaller in comparison. When increasing the area of the SLCF mitigation, which mostly means emission reductions further south, the LT BC and OC concentrations show fairly little further decrease, while the RA concentrations begin to decrease noticeably.

As sulfate emissions are not very strongly affected by the SLCF mitigation, there are no big differences in concentration changes between the different scenarios (Fig. 3f and i), and most of the visible changes are due to the CLE emission changes. It is, however, noteworthy that SU concentrations react differently to the CLE emission changes than BC and OC do. While LT SU concentrations decrease, RA concentrations increase. This happens because in the model SU is mainly emitted as $SO_2$, which then is chemically processed to form $SO_4$ and finally partitions to the aerosol phase via new particle formation (NPF) and condensation to pre-existing particles. Note here that with NPF we denote the formation of new particles through nucleation of $SO_4$ and their concurrent growth to CCN sizes through further condensation of $SO_4$ onto these particles (Kerminen et al., 2018; Lee et al., 2019). Any shift in aerosol concentrations alters the condensation sink for SU and thus may affect the horizontal and vertical locations of NPF. With a cleaner LT, more gaseous SU finds its way to the RA to undergo NPF there.

The RA increase in SU concentrations is larger in 2050 than in 2030, because of increased SU emissions at lower latitudes.

Figure 4 shows the yearly average Arctic column burdens of BC, OC, and SU. Based on the earlier defined boundary between lower troposphere (LT) and rest of the atmosphere (RA) in the Arctic (approximately 450 hPa), we computed separate RA (Fig. 4a–c), LT (Fig. 4d–f), and total (Fig. 4g–i) column mass burdens for BC, OC, and SU, respectively. As may be anticipated from the global emissions, the SU mass burdens vary very little between the different scenarios and mainly follow the changes in time of the CLE scenario. The BC burdens show the strongest relative changes between the scenarios, while the relative OC burden changes are much smaller than the BC burden changes, but still larger than the SU burden changes. In the following we will analyse the different species separately.

### 3.2.1 Black carbon

In the CLE scenario, the yearly LT BC burden decreases by $11.7 \pm 3.5\%$ ($6.1\,\mu g\,m^{-2}$) and $9.6 \pm 3.9\%$ ($4.9\,\mu g\,m^{-2}$) for 2030 and 2050, respectively, compared to 2010. In contrast, the current legislation together with the SLCF reductions in the Arctic Council member states (AC scenario) reduce the LT BC burden of 2010 by $39.4 \pm 3.9\%$ ($20.4\,\mu g\,m^{-2}$) and $43.4 \pm 3.6\%$ ($22.5\,\mu g\,m^{-2}$) for 2030 and 2050, respectively. This means that the influence of BC emissions in the regions close to the Arctic on Arctic LT BC burdens is substantial. For instance, in 2030, the difference in global BC emissions between the CLE and AC scenarios is only $-3.7\%$, while the difference in yearly average LT BC burden is $-31.3\%$, which amounts to $14.3\,\mu g\,m^{-2}$. Comparing this to a global SLCF reduction, the BC emissions in the same year decrease by $71.4\%$ between the CLE and GLOB scenarios, while the yearly average LT BC burden decreases by $57.5\%$ ($26.3\,\mu g\,m^{-2}$). The AC_ACT and AC_ALL scenarios only induce small reductions in LT BC burdens compared to the differences between CLE, AC, and GLOB. Expressing this

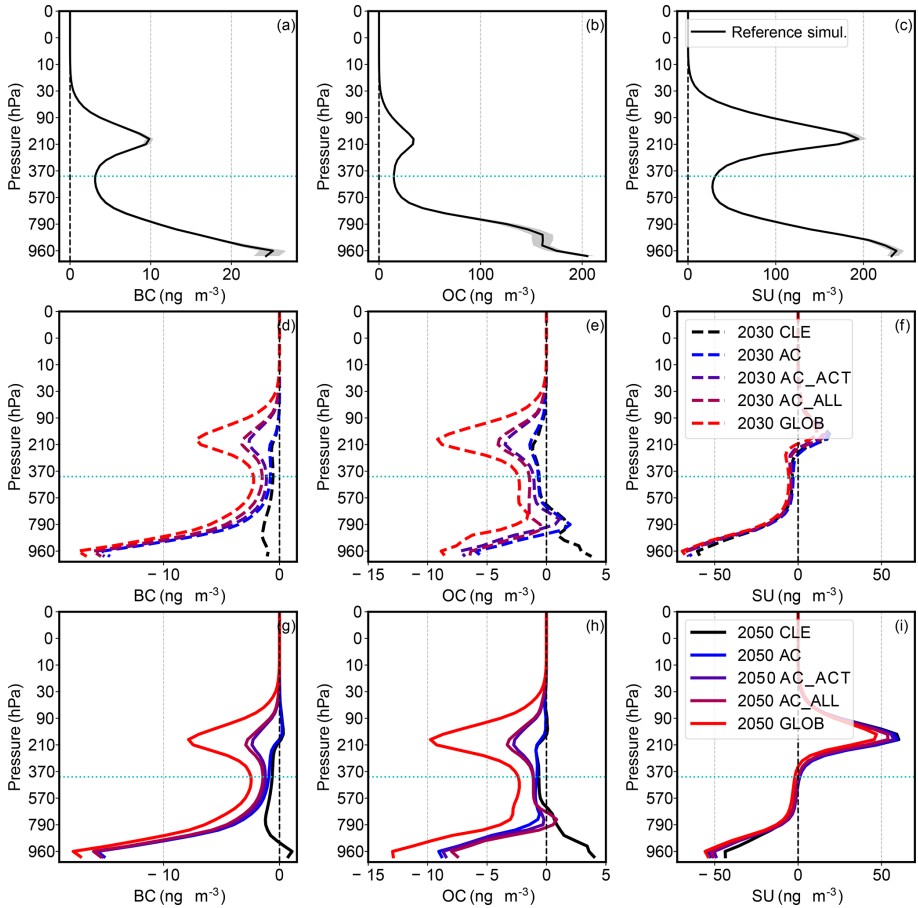

**Figure 3.** Arctic vertical profiles **(a–c)** in 2010 and their respective changes in 2030 **(d–f)** and 2050 **(g–i)** of BC **(a, d, g)**, OC **(b, e, h)**, and SU **(c, f, i)**. The grey shading in **(a)**–**(c)** denotes the interval between the 10th and 90th percentiles of the data.

in terms of a burden reduction efficiency, which one may define as the ratio between the relative Arctic BC mass burden reduction and the relative global BC emission reduction, this would result in a LT burden reduction efficiency of 8.4 for the AC scenario and 0.8 for the GLOB scenario. This is an important result: while the potential in BC emission reductions in the Arctic Council member states may be small compared to the global total, the potential to decrease Arctic BC concentrations close to the surface is substantial.

Doing the same analysis for the yearly average Arctic RA BC burdens, the decreases in the CLE and AC scenarios since 2010 are fairly small ($6.1 \pm 8.6$ % and $7.6 \pm 8.9$ %, respectively, for 2030). In fact, the variability in the change is larger than the actual simulated difference itself. Compared to the CLE scenario, the 2030 difference in RA BC burden in the AC scenario is $-1.6$ % ($1.1 \mu g\,m^{-2}$), and in the GLOB scenario it is $-71.5$ % ($49.9 \mu g\,m^{-2}$). AC_ACT reduces the RA BC burden by $15.2 \mu g\,m^{-2}$ from AC, while AC_ALL makes only a small additional contribution. Using the same definition as above, the RA burden reduction efficiency is 0.3 for the AC scenario and 1.0 for the GLOB scenario. This means that emissions further south have a higher relative im-

pact on RA BC burdens than emissions close to the Arctic (note that even though the emission reduction regions in the different scenarios do not strictly expand north to south, the countries that add most of the emissions in each scenario are distributed that way).

In summary, Arctic BC burdens follow BC emission reductions very systematically: the lower the BC emissions, the lower the total Arctic BC burdens. However, emission sources close to the Arctic affect BC burdens in the lower troposphere much more strongly than emission sources further south. The opposite is true for BC burdens in the rest of the atmosphere. As anthropogenic BC emissions at high latitudes are highest during the winter months and lowest during the summer, reductions in LT BC burdens are also strongest during the winter. The total LT BC mass burden, however, is higher during the summer months (but lower in the surface layer). The RA BC burdens do not show any seasonal trend.

### 3.2.2 Organic carbon

The yearly mean trends in Arctic OC burdens due to reductions in anthropogenic emissions largely follow the BC

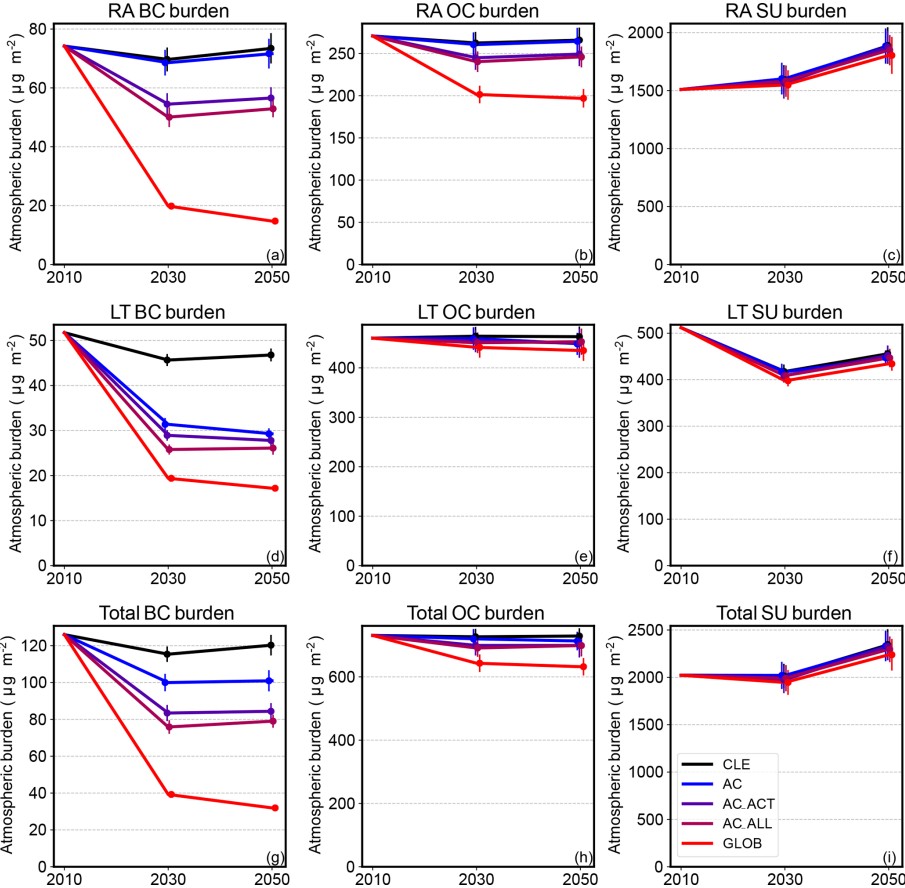

**Figure 4.** Arctic aerosol mass burdens for BC **(a, d, g)**, OC **(b, e, h)**, and SU **(c, f, i)**. Panels **(a)**–**(c)** show the rest of the atmosphere (RA) burdens, **(d)**–**(f)** show lower troposphere (LT) burdens, and **(g)**–**(i)** show the total column burden. The error bars show the standard deviation of the data.

burden trends. However, while the absolute values of the changes are of the same order of magnitude for OC and BC, the relative changes in the OC burden are much smaller. This is due to the high contribution from natural sources (e.g. biogenic sources and wildfires) to the total OC emissions, which is most noticeable in the LT during the summer. For instance, in 2010 the Arctic LT OC burden during the summer months is on average $1341.1\,\mu g\,m^{-2}$, which is about 15 times more than the LT BC burden during the same period. In contrast, the LT OC burden in the winter months is only $86.1\,\mu g\,m^{-2}$, which is only about 1.6 times higher than the LT BC burden. The seasonal variation in the RA OC burden is much less dramatic ($357.6$ and $226.8\,\mu g\,m^{-2}$ in the summer and winter, respectively).

Allowing the meteorology to evolve freely introduces considerable year-to-year variability in all transported species, because the wind and cloud fields have a relatively high natural variability, which affects both aerosol transport and removal. The variability in the mass burdens of any species is proportional to the total burden (due to both anthropogenic and natural emissions). Because the emission reductions considered here only affect the anthropogenic contribution to the total burden, which for OC was already quite small to start with, this natural variability introduces fairly large uncertainties into the changes in atmospheric OC burdens. For instance, in the CLE scenario, the change in Arctic LT OC burden from 2010 to 2030 during the summer months is $+3.0\pm8.1\,\%$, which equals $+40.3\pm109.0\,\mu g\,m^{-2}$. In other words, the natural variability is more than twice the actual average change and more than the total LT BC burden. Relative to the CLE scenario, the changes in Arctic LT OC summer burden in 2030 due to BC mitigation amount to $+0.7\,\%$ ($+10.2\,\mu g\,m^{-2}$) and $-1.7\,\%$ ($-23.1\,\mu g\,m^{-2}$) for the AC and GLOB scenarios, respectively. In the winter the corresponding changes are $-24.2\,\%$ ($-19.1\,\mu g\,m^{-2}$) and $-40.4\,\%$ ($-31.9\,\mu g\,m^{-2}$), respectively. For the RA the changes show even higher variability, but are probably climatically not as relevant, as will be discussed below.

### 3.2.3 Sulfate

The changes in SU burden are qualitatively quite different from the changes in BC and OC burdens. Like BC and OC,

the Arctic LT SU burden decreases in 2030 and 2050 compared to 2010, with the decrease in 2030 (about $18.4 \pm 4.0\%$ for CLE) being larger than in 2050 ($10.9 \pm 4.3\%$ for CLE). In the RA, on the other hand, the SU burden increases by $5.8 \pm 10.9\%$ and $25.1 \pm 12.7\%$ in 2030 and 2050, respectively, for the CLE scenario. This stronger increase in RA SU burden in 2050 compared to 2030 aligns well with the strong increases in SU emissions in India in 2050.

In general, the Arctic SU burden is largely unaffected by the SLCF reduction scenarios and appears to be dominated by the emission changes in the CLE scenario. However, there appears to be a slight trend of both the LT and RA SU burdens being lower with globally increasing coverage of the SLCF emission reductions. The uncertainties in these values, however, are very big, and especially the differences between the changes in different scenarios are much smaller than the accompanying uncertainties.

As has been discussed in the OC section above, the larger relative variability in SU burden changes can be explained by the relatively small change in total emission strength. While the anthropogenic SU emission reduction is the main driver of the average change in SU mass burden, the total SU emission strength (anthropogenic and natural) directly influences the magnitude of the variability of the change. Furthermore, both SU and OC have a higher water solubility than BC, which makes these substances more susceptible to aerosol–cloud interactions and wet removal, both processes being highly variable in ECHAM-SALSA.

## 3.3 Aerosol–cloud interactions

The discussed changes in aerosol concentrations have relatively strong effects on the cloud properties. Figure 5a shows the vertical distribution of the average concentration of aerosol particles with diameters larger than 100 nm ($N_{100}$) over the Arctic for the 2010 reference simulation. $N_{100}$ is a commonly used proxy for the concentration of cloud condensation nuclei (CCN) (Dusek et al., 2006; Janssen et al., 2011; Tröstl et al., 2016). Figure 5d and g show the changes in the $N_{100}$ profiles for 2030 and 2050, respectively. Here the influence of all aerosol species on $N_{100}$ can clearly be seen. In the RA, the $N_{100}$ trend appears to be dictated mainly by the SU trends. This indicates that the changes in RA $N_{100}$ are mainly caused by changes in new particle formation (NPF). In the LT, on the other hand, the $N_{100}$ trends appear to be more dependent on the BC and OC trends, especially close to the surface. This indicates that here changes in $N_{100}$ are mainly governed by primary emissions. In the CLE scenario, the LT $N_{100}$ burden decreases by $4.5 \pm 5.5\%$ and $2.9 \pm 5.7\%$ for 2030 and 2050, respectively, compared to 2010. For the AC scenario, the LT $N_{100}$ burden decreases by $13.2 \pm 5.5\%$ and $12.6 \pm 5.5\%$ for the same years. As for the BC and OC mass burden trends, these changes are more pronounced during the winter months and are least distinguishable during the summer.

Figure 5b shows the yearly average vertical distribution of the Arctic cloud droplet number concentration (CDNC) for the 2010 reference simulation, while Fig. 5e and h show the respective changes for 2030 and 2050. Since most water clouds occur in the lower part of the atmosphere, we will restrict this discussion to LT CDNC. As may be expected, the changes in $N_{100}$ affect the Arctic CDNC values. The change in LT CDNC burden since 2010 in the CLE scenario is fairly small: $-3.5 \pm 3.6\%$ in 2030 and $-1.2 \pm 3.4\%$ in 2050. The change in CDNC burden from the CLE to the AC scenario is $-7.9\%$ and $-10.0\%$ for 2030 and 2050, respectively, and varies only slightly for the other scenarios. This is in line with the LT OC and BC mass burdens changing most in the AC scenario, while the other SLCF scenarios affect the RA mass burdens more. Similarly to the OC and BC mass burdens and the $N_{100}$ burdens, CDNC burden changes are strongest during the winter and weakest during the summer.

A decrease in CDNC means that the cloud droplets are on average larger, which renders the clouds less reflective, amounting to a net warming effect (Twomey, 1977). For the scattering SU aerosols this means that the direct warming effects of SU reductions are amplified by the decrease in CDNC, while for the absorbing BC the reduction in CDNC counteracts the cooling effect of BC reductions. On the other hand, smaller CDNC values may accelerate precipitation formation, which in turn may shorten the cloud lifetime (Albrecht, 1989), which may reduce the cloud fraction. A reduced cloud fraction, depending on the conditions, may have either a warming or cooling effect. On the one hand, if the cloud fraction is smaller, less sunlight is reflected back to space, which nets to a warming of the atmosphere. On the other hand, less outgoing longwave radiation is reflected back to the surface, which nets to a cooling of the surface. The changes in Arctic cloud fraction will be discussed below in combination with the radiative forcings.

Figure 5c, f, and i indicate a small upward shift in the Arctic cloud cover vertical profile: the cloud cover fraction decreases in the LT and increases in the RA (this can also be seen in Figs. S5 and S6 in the Supplement). However, these shifts are statistically not significant, and we therefore do not investigate this further.

## 3.4 Radiative forcing

### 3.4.1 Global values

The global aerosol all-sky short-wave (SW) direct radiative forcing ($sRF_A$) is shown in Fig. 6a. The aerosol long-wave (LW) direct radiative forcing is an order of magnitude smaller than the SW forcing and will therefore not be discussed here. Omitting a detailed quantitative analysis, it can be seen that the $sRF_A$ values very well reflect the emission reductions of the different scenarios: the more the BC emissions are reduced in any particular year, the larger the global cooling effect that can be seen in the $sRF_A$ values. A no-

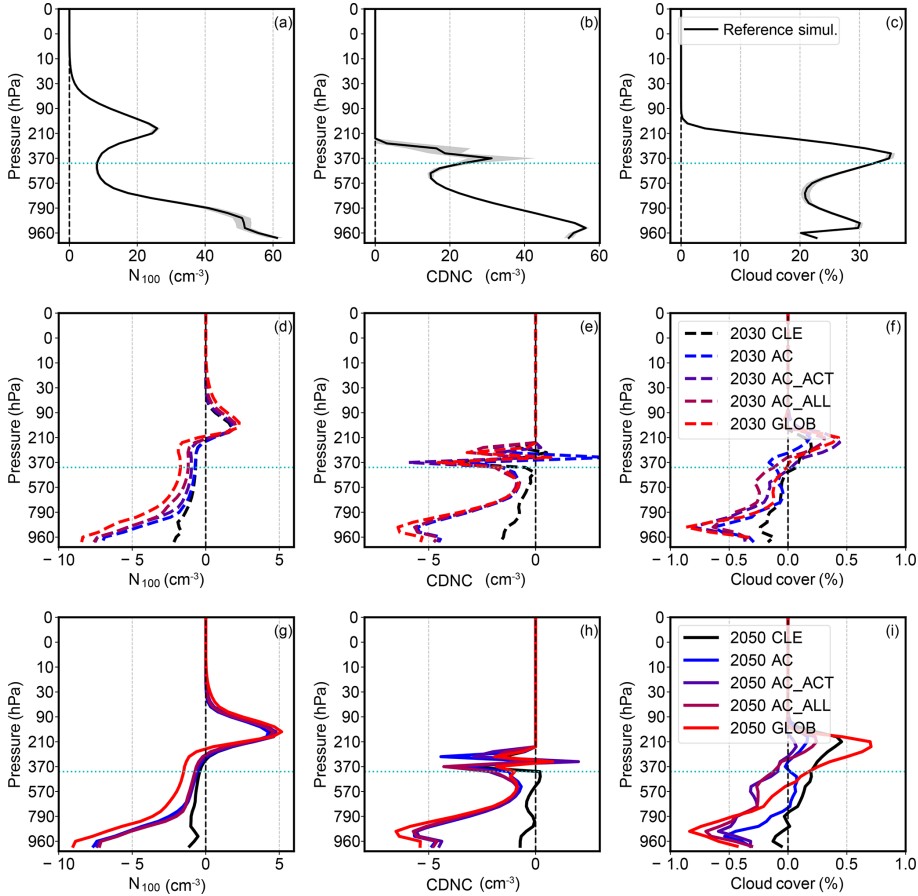

**Figure 5.** Arctic vertical profiles **(a–c)** in 2010 and their respective changes in 2030 **(d–f)** and 2050 **(g–i)** for $N_{100}$ **(a, d, g)**, CDNC **(b, e, h)**, and cloud cover fraction **(c, f, i)**. The grey shading in panels **(a)**–**(c)** denotes the interval between the 10th and 90th percentiles of the data.

table feature of the BC direct radiative forcing is that the BC all-sky forcing is typically larger (more negative) than the clear-sky forcing (not shown), which is opposite to the direct radiative forcings of scattering aerosols. This occurs because in a cloudy sky more sunlight is reflected back to space and thus the absolute amount of short-wave radiation leaving the planet is larger, which amplifies the light absorption effect of BC (Samset and Myhre, 2011, 2015; Kühn et al., 2014).

Compared to the $sRF_A$ values of the SLCF scenarios, the $sRF_A$ that is caused by the reductions in SU emissions in the CLE scenario is much smaller in magnitude. This can be seen by following the black line for the CLE scenario in Fig. 6a: between 2010 and 2030, where global SU emissions decrease, the global $sRF_A$ is slightly positive ($0.03 \pm 0.07$ W m$^{-2}$), while between 2030 and 2050, where global SU emissions recover, the $sRF_A$ is slightly negative ($-0.05 \pm 0.08$ W m$^{-2}$). Altogether, a global implementation of the maximum feasible BC emission reductions (scenario GLOB) produces $sRF_A$ values of $-0.45 \pm 0.08$ and $-0.57 \pm 0.07$ W m$^{-2}$ for 2030 and 2050, respectively, relative to the 2010 reference scenario. The $sRF_A$ values for the other scenarios follow the BC emission amounts fairly well.

However, when also taking the indirect effects into account, the picture changes quite dramatically.

Figure 6b shows the total global short-wave radiative forcing ($sRF_{TOT}$; this quantity also includes rapid adjustments of the atmosphere to the changing aerosol emissions) at the top of the atmosphere. First off, the reduction in SU emissions in the CLE scenario between 2010 and 2030 produces a noticeable, statistically significant warming signal of $0.37 \pm 0.31$ W m$^{-2}$, which decreases to $0.16 \pm 0.28$ W m$^{-2}$ in 2050, where the SU emissions almost recover to the value of 2010. On the other hand, the reductions in BC and OC in the SLCP scenarios compared to CLE indicate, depending on the scenario, either cooling or warming, with no visible systematic response to the amount of BC and OC reduction. However, the variability in $sRF_{TOT}$ is considerable, and the differences between different scenarios are statistically not significant. As the $sRF_A$ values clearly are negative, this change in sign of the radiative forcing must be due to changes in planetary albedo, which may be attributed to either surface changes or changes in clouds. On the global average, the surface albedo (not shown) varies only slightly and may contribute to the variability in the radiative forcings,

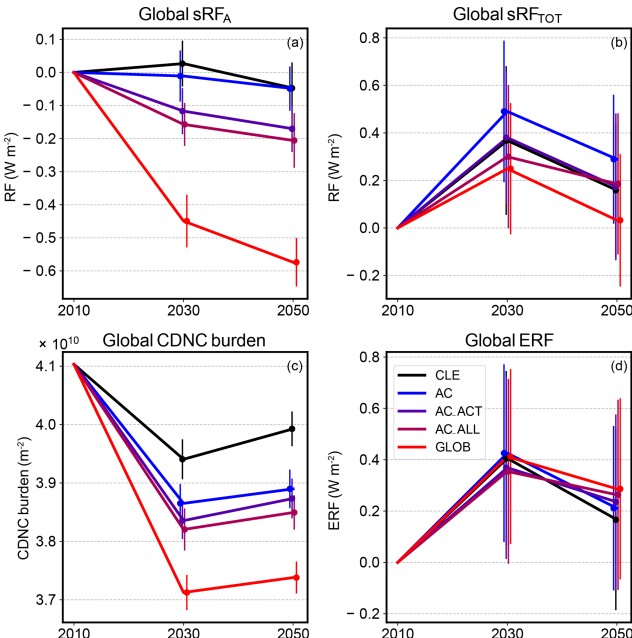

**Figure 6.** Global **(a)** direct short-wave aerosol radiative forcing, **(b)** total short-wave radiative forcing, **(c)** CDNC burden, and **(d)** effective radiative forcing. The error bars show the standard deviation of the data.

but cannot explain the difference between aerosol and total short-wave RF. We find, however, a strong effect in global average CDNC burdens (Fig. 6c). The amount of global average CDNC decreases very systematically with the amount of reduction in aerosol emissions. Here the effects of both SU reduction in the CLE scenario and BC and OC reduction in the SLCF scenarios are clearly visible. Between 2010 and 2030, the global average CDNC burden decreases by $3.5 \pm 3.3\%$ from $3.7 \times 10^{10}$ to $3.6 \times 10^{10}$ m$^{-2}$ in the CLE scenario, and the maximum reduction in CDNC due to SLCF reductions is $12.9 \pm 2.9\%$ TS1 in 2030 between the CLE and GLOB scenarios. The relation between CDNC and cloud radiative forcing has been studied in Storelvmo et al. (2009). They list the radiative forcings due to increases in CDNC at 950 hPa for different models. Interestingly, even though the CDNC base values and the CDNC increases vary a lot between models, the resulting radiative forcings are within 0.62 and 1.94 W m$^{-2}$. Using the values provided in Table 1 in Storelvmo et al. (2009), one can calculate a linear relation between percental CDNC change and cloud radiative forcing. The values range from $-0.14$ cm$^3$ W m$^{-2}$ for very low CDNC base values (41.9 cm$^{-3}$) to $-0.01$ cm$^3$ W m$^{-2}$ for very high CDNC base values (158.7 cm$^{-3}$). In the simulations performed here, we find an average cloud-weighted CDNC at 940 hPa of 59.3 cm$^{-3}$ with percental changes of up to 11.5 % in the SLCF mitigation scenarios. For these values, the difference between sRF$_A$ and sRF$_{TOT}$ that we observe can be explained with a factor between $-0.05$ and

$-0.06$ cm$^3$ W m$^{-2}$, which lies well in the range derived from Storelvmo et al. (2009). In addition to their ability to act as CCN, BC particles can also affect clouds through several semi-direct effects. As BC inside or close to clouds absorbs radiation, it can affect cloud droplet evaporation and atmospheric stability, which can also affect cloud properties and lead to cooling (Koch and Del Genio, 2010). These effects, however, are very difficult to distinguish from each other in ECHAM-HAMMOZ and are therefore not further diagnosed here.

There is also a decrease in cloud time fraction (fraction of the total simulated time that a grid box is in cloud) with decreasing aerosol emissions (not shown). The effect is fairly small, but the cloud time fraction does decrease systematically with the decreasing strength in aerosol emissions. In our simulations the decrease in cloud time fraction is most noticeable between the CLE and AC scenarios and between the AC_ALL and GLOB scenarios (the differences between AC, AC_act, and AC_all are very small). This implies that decreasing aerosol emissions close to very pristine environments have the biggest effect on cloud time fraction. Generally a reduced cloud time fraction is attributed a warming effect, because the amount of reflected sunlight back to space is reduced (Albrecht, 1989). However, at the same time the clouds also reflect less long-wave radiation back to the surface (Ramanathan et al., 1989), which makes the effect on radiative forcings of this phenomenon less predictable. However, as the maximum change in cloud time fraction is only of the order of about 1 %, we reason that most of the effect of aerosol–cloud interactions on the radiative forcings is through the aerosol effect on CDNC.

Figure 6d shows the global effective radiative forcing (ERF), which is calculated as the difference in net radiation budget (SW and LW) at the top of the atmosphere (TOA) between the simulation scenarios and the 2010 reference scenario. For the CLE scenario the ERF is positive, amounting to 0.41 and 0.29 W m$^{-2}$ TS2 for 2030 and 2050, respectively, which is in line with the sRF$_{TOT}$ values. This reflects quite well the global SU emission reductions, allowing for shifts in the location of the emissions. The differences between the different SLCF scenarios are very small and accompanied by very large uncertainties. This means that globally the direct radiative effects due to the BC emission reductions are counteracted by cloud effects. This has also been reported in previous studies (Kuwata et al., 2009; Koch and Del Genio, 2010; Smith et al., 2018).

### 3.4.2 Arctic values

Figure 7a, d, and g show the total Arctic aerosol direct radiative forcing (RF$_A$, which includes both SW and LW radiation; note the difference to sRF$_A$ discussed for the global forcings, which considers only SW radiation) for winter, summer, and all year, respectively. Like for the global values, the Arctic RF$_A$ follows the BC burden reduction amounts

very systematically, with maximum $RF_A$ values of $-0.39$ and $-0.44\,W\,m^{-2}$ for 2030 and 2050, respectively, for the GLOB scenario. In a cloudy sky the RA BC concentrations contribute much more strongly to the direct radiative forcing than the LT values (all BC mass below cloud is screened) (Samset and Myhre, 2011; Kühn et al., 2014; Samset and Myhre, 2015). This is why the BC emission reductions of the Arctic Council member states have a relatively small effect on the $RF_A$, which mostly cause lower-level changes in BC concentrations. Because of the strong seasonal cycle of solar insolation, the Arctic $RF_A$ is much stronger during the summer than during the winter.

The Arctic ERF values (Fig. 7b, e, and h) principally follow the results obtained for the global ERF. However, here the relative uncertainty is even larger, taking values of the order of $\pm 2\,W\,m^{-2}$. The ERF differences between the different scenarios are as high as $0.5\,W\,m^{-2}$, with no systematic ordering concerning the BC emission reduction strengths of the SLCF mitigation scenarios. Furthermore, the Arctic ERF values appear to be dictated quite strongly by the Arctic cloud cover (Fig. 7c, f, and i). In particular, the winter ERF and cloud cover have a strong positive correlation (0.80), while the summer ERF and cloud cover have a strong negative correlation ($-0.97$). This occurs because during the winter, the long-wave warming of clouds dominates, while during the summer it is the short-wave cooling effect. In the yearly average, the correlation between ERF and cloud cover is much weaker ($-0.64$), because both short- and long-wave effects are important. This is also visible when comparing Fig. 7h and i.

The reason why $RF_A$ and ERF are so different in the Arctic can be explained by cloud changes. During the winter, when BC and OC emission reductions in the AC scenario are largest, there is also a very strong decrease in LT CDNC burden (not shown) between the CLE and AC scenarios (21.4 %), while the differences between AC and the other SLCF mitigation scenarios are small, but systematic. During the summer the CDNC trends for the different scenarios are similar, but much less pronounced. Like for the global radiative forcings, we interpret the changes in Arctic CDNC as the main driver of the differences between $RF_A$ and ERF.

As already mentioned, the Arctic ERF values show large uncertainties. The main contributions to these uncertainties are the strong natural variability in Arctic cloud cover and yearly average surface albedo, the latter of which is due to the year-to-year variability in snow cover. Other possible contributors are the variability in aerosol burdens, CDNC, and heat transport into the Arctic. Equally strong uncertainties in ERF have also been observed elsewhere (e.g. Cherian et al., 2017).

## 3.5    Surface

Deposition of BC on ice and snow is widely reported to strongly affect the surface albedo and accelerate snowmelt and ice melt (Quinn et al., 2011; Bond et al., 2013; Sand et al., 2016). In ECHAM-HAMMOZ the deposition of BC is separated into wet deposition, dry deposition, and sedimentation, with wet deposition making the largest contribution to the total. As dry deposition is a function of the BC concentrations close to the ground and wet deposition only depends on in- and below-cloud BC concentrations, it is the LT BC concentrations that dictate the BC deposition rates in the Arctic. This can clearly be seen in Fig. 8: the LT BC burden reductions of the SLCF scenarios are directly reflected in the BC deposition rates. Even though the SLCF emission reductions in the Arctic Council member states, compared to the CLE scenario, only comprise 5.2 % and 5.3 % of the globally feasible total reductions, these reductions can reduce the Arctic BC deposition by 29.3 % and 33.8 % in 2030 and 2050, respectively. This comprises 57.8 % and 59.7 % of the achievable decrease in Arctic BC deposition for global implementation of the SLCF emission reductions.

The Arctic surface albedo varies strongly during one year due to the big changes in snow and ice cover extent. Because the time that an area is covered by snow during one winter can change considerably from one year to another, the variations in yearly Arctic surface albedo are big as well (Fig. 9a). Surface albedo can take values between 0 and 1, and a change of 0.001 in surface albedo amounts to a change of $0.1\,W\,m^{-2}$ of sunlight absorption per $100\,W\,m^{-2}$ of solar insolation. As can be seen in Fig. 9a, the differences in average Arctic surface albedo between the scenarios can be up to 0.0012, with standard deviations of the order of 0.003. Note that in the simulations performed here the monthly varying sea ice extent was the same in each simulation and in each simulation year, which means that the variation in surface albedo is caused only by the varying snow cover.

Like many other climate models, ECHAM-HAMMOZ does not account for changes in snow albedo due to the deposition of absorbing aerosols. The effect of the modelled BC deposition flux on the Arctic snow albedo has been approximated for a wide selection of AeroCom (Aerosol Comparisons between Observations and Models) models in Jiao et al. (2014), which also included a previous version of the model used here, ECHAM5-HAMMOZ. There the modelled Arctic BC deposition fluxes were used as input to an independent surface model, which then computed the radiative effect due to the computed changes in surface albedo. In order to approximate the radiative forcings due to the changes in the Arctic BC deposition flux in this study, we used the data provided in Jiao et al. (2014) of all modelled Arctic BC deposition fluxes and the resulting radiative effects to derive a linear relationship between the two quantities. To this end, we performed an orthogonal distance regression (ODR) using the relationship $RE = a \times F_{BC} + b$, where RE is the Arctic radiative effect in $W\,m^{-2}$, $F_{BC}$ is the yearly average Arctic BC deposition flux in $kg\,yr^{-1}$, and the coefficients $a$ and $b$ have the values $9.76 \pm 1.05 \times 10^{-10}\,W\,m^{-2}\,(kg\,yr^{-1})^{-1}$ and $3.69 \pm 2.21 \times 10^{-2}\,W\,m^{-2}$, respectively. The regression

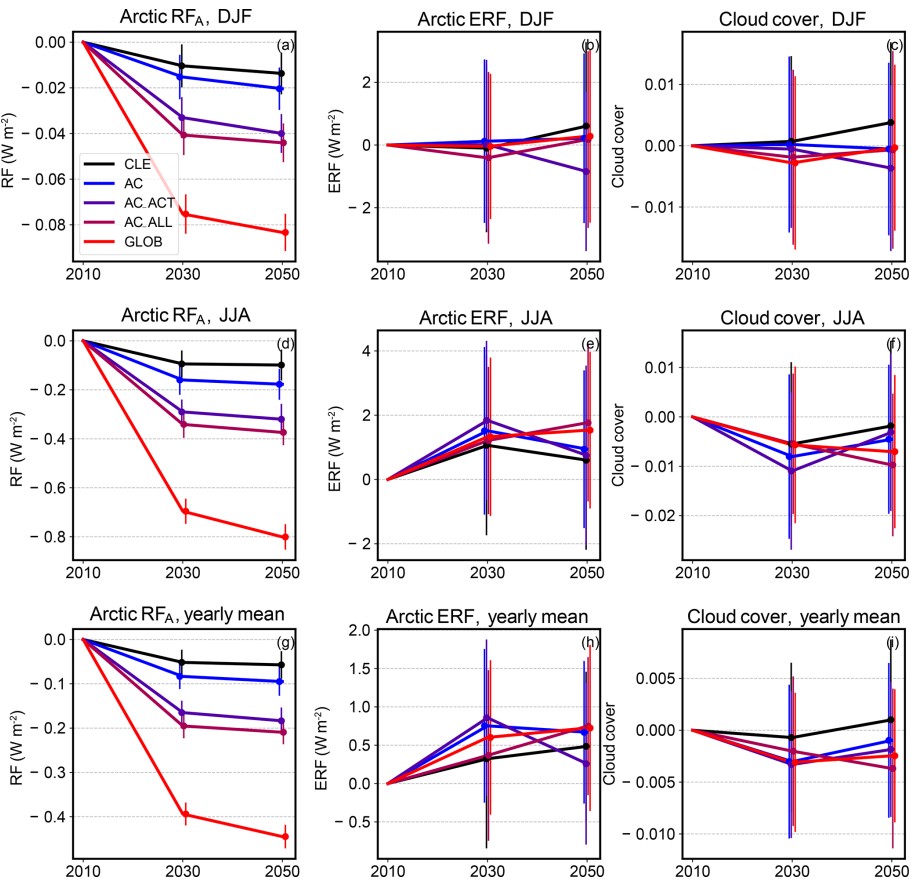

**Figure 7.** Arctic total aerosol all-sky direct radiative forcing **(a, d, g)**, effective radiative forcing **(b, e, h)**, and cloud cover **(c, f, i)** for winter **(a–c)**, summer **(d–f)**, and yearly **(g–i)** averages. The error bars show the standard deviation of the data.

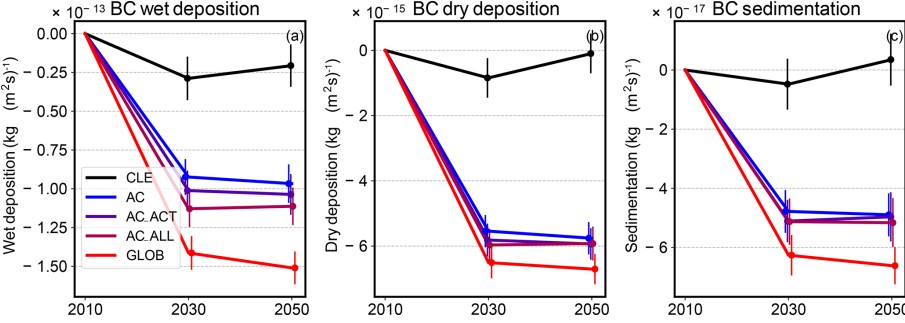

**Figure 8.** Arctic BC deposition fluxes: **(a)** wet deposition, **(b)** dry deposition, and **(c)** sedimentation. The error bars show the standard deviation of the data.

is visualised in Fig. 9b. The fit shows that the relation between BC deposition rate and RE is remarkably linear, with relatively small standard deviations, even though we here used only yearly average fluxes, while Jiao et al. (2014) used monthly two-dimensionally resolved deposition fluxes as model input. This may, however, mainly be due to the fact that Jiao et al. (2014) used the same nudged meteorology for all simulations and only changed the prescribed BC deposition fluxes according to the models tested. The yearly Arctic

BC deposition rates simulated here are well within the data range of the models used in Jiao et al. (2014).

Using the linear relation derived here, we computed the Arctic radiative forcings due to BC deposition ($RF_{snow}$) on snow and ice that may be expected for the simulations performed here (Fig. 9c). As may have been expected, the $RF_{snow}$ values follow the BC wet deposition values ($F_{BC,wet}$) (Fig. 8a) very well, because of the linear relationship between $F_{BC}$ and $RF_{snow}$ and because wet deposition is the

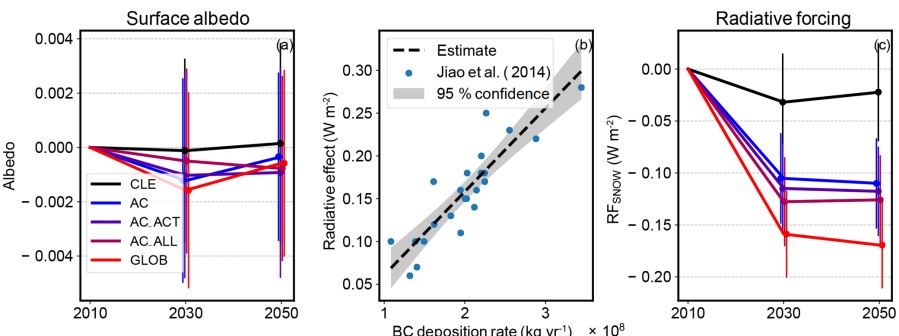

**Figure 9. (a)** Yearly average Arctic surface albedo, **(b)** linear regression to data from Jiao et al. (2014), and **(c)** Arctic surface radiative forcing due to changes in BC deposition using results from **(b)**. The error bars show the standard deviation of the data.

dominant BC deposition process. As emissions close to the Arctic contribute most to the LT BC burdens and thus have the biggest impact on Arctic BC deposition fluxes, these emissions also have a big influence on the $RF_{snow}$ values. For instance, in 2030 $RF_{snow}$ between CLE and AC is $0.07 \pm 0.04 \, W \, m^{-2}$, more than half of the value obtained between CLE and GLOB, $0.13 \pm 0.05 \, W \, m^{-2}$. For any given year, the absolute value of $RF_{snow}$ increases with the amount of BC emission reduction, but the differences between AC, AC_ACT, and AC_ALL are fairly small. Considering the relatively small amount in BC emission reduction in AC compared to the other scenarios, the contribution of the BC mitigation of the Arctic Council member states to the Arctic $RF_{snow}$ is quite substantial.

Jiao et al. (2014) remark that in the AeroCom models investigated there, the BC emission fluxes were constant in time, which is unrealistic, because BC emissions in some sectors (e.g. residential combustion and energy production) are higher during the winter and lower during the summer, especially at higher latitudes. According to them, this most likely leads to an underestimation of $RF_{snow}$, because less BC is deposited during the winter. As the albedo changes affect $RF_{snow}$ most during the spring time, the winter is the most important period with respect to deposited BC affecting snow albedo. In the simulations performed in this study, we used the ECLIPSE emission scenarios, which provide a more realistic annual distribution of the emissions. Therefore the values for $RF_{snow}$ would most likely be larger (in magnitude) if similar simulations had been performed using the BC deposition fluxes modelled here. On the other hand, using monthly average BC deposition fluxes instead of deposition that is simultaneous with precipitation may overestimate $RF_{snow}$ (Doherty et al., 2014). Altogether, to produce better estimates of $RF_{snow}$, an online snow and ice albedo model that accounts for BC deposition should be included. However, considering the large natural variability in snow cover (and thereby in surface albedo) in simulations with freely evolving meteorology, it may be equally challenging to extract Arctic RF values of that magnitude from such simulations as it is for RF due to atmospheric aerosol changes.

Comparing the Arctic $RF_{snow}$ values to the Arctic ERF values, the latter of which differ at maximum by $0.4 \, W \, m^{-2}$ and have standard deviations of the order of $0.5 \, W \, m^{-2}$ TS3, the $RF_{snow}$ values are relatively small. In fact, adding the surface snow albedo effect to the atmospheric ERF does not help to separate the total forcings into a meaningful or systematic order (not shown).

## 3.6 Human health

We used the number of premature deaths due to elevated concentrations of particulate matter as an indicator of the health benefits that can be achieved due to the emission reductions in the different scenarios. The health benefits of the emission reduction of each scenario have been computed for each of the four regions defined in Fig. 1. Thereby we found that, compared to the current legislation (scenario CLE), emission reductions in the Arctic Council member states alone (scenario AC) reduce the number of premature deaths by 30 000 (19 %) and 41 055 (23 %) in the Arctic Council member states in 2030 and 2050, respectively (Fig. S1). The additional health benefits outside the Arctic Council member states are relatively small. Globally the emission reductions in the Arctic Council member states prevent 33 000 and 47 000 premature deaths in 2030 and 2050, respectively (Fig. S4). In general, it can be said that the health benefits of the emission reductions are always largest in the region where the emissions are actually reduced (Figs. S1–S4). For instance, reducing emissions in the active observer states in addition to the Arctic Council member states increases the number of prevented deaths within the active observer states from 43 000 to 206 000 in 2050. This means that even regions that do not directly benefit from the impact on Arctic climate still have a strong motivation to reduce their SLCF emissions. Globally, 329 000 (9 %) premature deaths could be avoided in 2030 if the Arctic Council member states and all observer states implement all SLCF mitigation options, which is 18 % less (403 000 (11 %) avoided deaths) than for a full global implementation.

These estimates are smaller than, for example, in Anenberg et al. (2012), who estimated, for a similar global mitigation scenario to the GLOB scenario used here, that full implementation could annually avoid 0.6–4.4 million premature deaths globally in 2030. We acknowledge that the overall particulate matter (PM) concentrations also contain other species, e.g. ash and secondary material from atmospheric transport, which is why this estimate is a conservative one and should be seen as a demonstration of the magnitude of the effects rather than a full analysis of PM-related health effects. Furthermore, due to the coarseness of the models used here (and global models in general), concentration spikes (both spatial and temporal) cannot be simulated to their full extent, which lessens the overall impact of PM concentrations on human health. Another reason for the discrepancy is probably the different exposure–response function used, which in TM5-FASST flattens off at higher $PM_{2.5}$ concentrations. However, as all these shortcomings are true for both the reference scenario (CLE) and the mitigation scenarios, the relative changes in premature deaths contain valuable information nonetheless.

## 4 Conclusions

In order to assess the impacts of black carbon (BC) mitigation policies on Arctic climate, we studied the radiative forcings that occur when such policies are applied. To this end, we constructed emission scenarios using the fairly recently published ECLIPSE v5a emission scenarios (Stohl et al., 2015; Klimont et al., 2017). The scenarios were constructed such that they reflected the full implementation of all currently agreed policies and furthermore implemented the maximum feasible BC mitigation in a successively increasing area of the globe, including the Arctic Council member states, active observer states, all observer states, and finally the entire globe. The different geographical extents for mitigation were studied because of the extensive work that has already been done by the Arctic Council regarding BC mitigation measures and the large interest in its member states to actually reduce BC emissions. The probability that the emission reductions will be implemented in part or all of the areas defined in Fig. 1 is therefore relatively high. Thus studying these scenarios is very timely and important. The scenarios account for the simultaneous decrease in co-emitted species. We restricted this study to the radiative forcings due to changes in aerosol emissions. In particular, greenhouse gas concentrations were the same in all simulations.

We found a very strong relation between total global reduction in anthropogenic BC emissions and Arctic BC mass burdens. Similar relations were also found for organic carbon (OC) and sulfate (SU), but for these species the natural background is much larger and thus the changes in Arctic burden are of less relative importance. As reported elsewhere (Stohl, 2006; Quinn et al., 2011), we find that emissions close to the

Arctic influence more the BC concentration near the surface, while emissions further south mainly control the BC concentrations at higher altitudes. We here divided the Arctic BC mass burdens into a lower troposphere (LT) and a rest of the atmosphere (RA) contribution and found that, even though the maximum feasible reductions in BC emissions in the Arctic Council member states are small compared to the global potential, the effects that these reductions have on the LT BC burdens are considerable. This is very important, because the LT BC burden has a very strong influence on Arctic BC deposition to the surface, while the RA BC burden affects BC deposition only slightly.

We find a fairly linear relationship between Arctic BC and OC burden and Arctic direct aerosol radiative forcing ($RF_A$), independent of the altitude at which the BC concentration changes. BC and OC are usually attributed opposite effects on direct radiative forcing and cannot really be separated here, because the changes in BC and OC burdens are so similar in the different scenarios. However, due to the extensive masking by clouds in the Arctic, the OC effect is expected to be much smaller than the BC effect. There is no discernible effect of SU on Arctic $RF_A$.

In contrast to the $RF_A$, the Arctic effective radiative forcing (ERF) (Lohmann et al., 2010; Forster et al., 2016) shows no noticeable trend as a function of BC or OC burden, and the ERF values are accompanied by very large uncertainties. We argue that the $RF_A$ contribution to the ERF is cancelled by changes in cloud droplet number concentrations (CDNC) and cloud cover (Twomey, 1977; Albrecht, 1989; Storelvmo et al., 2009). The uncertainties in the ERF are due to the strong natural variations in the model meteorology, which ultimately causes variations in CDNC, cloud cover, surface albedo, and, possibly, energy transport into the Arctic. Similar uncertainties in Arctic ERF have also been reported in other studies (Cherian et al., 2017). Our model does not account for snow albedo changes due to BC deposition. We therefore tried to estimate the resulting radiative forcing ($RF_{snow}$) in our simulations with the help of results from another study (Jiao et al., 2014). We found that the $RF_{snow}$ due to the simulated BC emission reductions may be relevant, but would still be small compared to the uncertainties in snow cover fraction and ERF that we encountered.

The potential of BC mitigation to achieve a slowing of Arctic warming on a relatively short timescale has been discussed widely in the literature (Quinn et al., 2011; Bond et al., 2013; Jiao et al., 2014; Samset and Myhre, 2015; AMAP, 2015; Cherian et al., 2017). BC is a good candidate for this very important goal because of its strong interaction with solar radiation. However, as was shown in this study, conclusions about the efficacy of BC mitigation measures cannot be based on the direct effects of BC–radiation interactions alone. Instead, co-emitted species and aerosol–cloud interactions also have to be taken into account. According to the fifth assessment report of the IPCC (Stocker et al., 2013), aerosol–cloud interactions contribute the largest

amount of uncertainty to radiative forcing estimates and climate projections in climate models. These uncertainties arise due to differences in different climate models, with ECHAM-HAMMOZ and especially ECHAM-SALSA having a stronger-than-average aerosol–cloud coupling (Smith et al., 2018). It may therefore well be that, using the same scenarios used in this study, another model would predict more cooling. The uncertainties in ERF reported here are due to model-internal variability. Here we used 30 integration years for our simulations, which is recommended for ERF values of at least $0.1\,\mathrm{W\,m^{-2}}$ (Forster et al., 2016). As the area studied here is relatively small and the Arctic surface albedo and cloud cover are highly variable already, it may be possible that much longer integration times are needed in order to obtain conclusive results, if the Arctic ERF is small. This may, however, be computationally too costly. Other methods to estimate the ERF have been suggested (Forster et al., 2016), which may reduce variability but often suppress important climate-relevant processes, like, for instance, the effects of changes in meteorological conditions on cloud dynamics (Forster et al., 2016; Zhang et al., 2014). It may therefore be necessary to develop alternative methods to quantify climate effects in the Arctic. Finally, estimating climate impacts by computing the ERF due to a given change in emissions will never draw the entire picture, because fixing the sea surface temperature (SST) and sea ice cover (SIC) prohibits important climate feedbacks, like e.g. changes in the oceanic heat transport into the Arctic and the resulting changes in SST and SIC, changes in precipitation, and accelerated snowmelt. All these feedbacks affect Arctic surface temperatures in addition to the ERF, and it may therefore be better to use a fully coupled ocean–aerosol–climate model to estimate Arctic temperature responses to changing aerosol emissions.

In addition to the climate impacts, reducing BC emissions also has positive effects on human health. Using the TM5-FASST model (Huijnen et al., 2010; Van Dingenen et al., 2018), we found that globally 329 000 and 402 000 premature deaths could be prevented by 2030 and 2050, respectively, if the proposed emission reductions (Stohl et al., 2015; Klimont et al., 2017) are fully implemented in all Arctic Council member and observer states. Compared to other studies (Anenberg et al., 2012), this is a conservative estimate, because we only considered part of all fine particulate matter ($PM_{2.5}$) in this study.

To conclude, even though the direct radiative effect of BC mitigation is easily quantifiable, the accompanying aerosol–cloud interactions of BC and its co-emitted species are still highly uncertain. Together with the natural variability of surface albedo and meteorology, this makes the overall effect on Arctic climate hard to assess. This does, however, not mean that BC mitigation is not useful in slowing Arctic warming, especially considering that several climate feedbacks that are not considered here (e.g. snow albedo feedback and BC effects on cloud dynamics) may further increase the BC warm-

ing potential. Further studies, including more models, will be needed in order to obtain higher-confidence estimates of the efficacy of BC mitigation strategies.

*Code availability.* The ECHAM6-HAMMOZ model is made available to the scientific community under the HAMMOZ Software Licence Agreement, which defines the conditions under which the model can be used. The licence can be downloaded from https://redmine.hammoz.ethz.ch/attachments/download/291/License_ECHAM-HAMMOZ_June2012.pdf (HAMMOZ consortium, 2012).

*Data availability.* The model data can be reproduced using model revision r5888 from the repository https://redmine.hammoz.ethz.ch/projects/hammoz/repository/show/echam6-hammoz/branches/fmi/white (HAMMOZ consortium, 2019). Alternatively, the data can be obtained directly from the authors. The settings for the simulation are given in the same repository, in folder https://redmine.hammoz.ethz.ch/projects/hammoz/repository/show/echam6-hammoz/branches/fmi/acp_2019_09_24_settings (HAMMOZ consortium, 2019). The ECLIPSE emission input files are available from http://www.iiasa.ac.at/web/home/research/researchPrograms/air/ECLIPSEv5a.html (IIASA, 2015). All other emission input files are ECHAM-HAMMOZ standard and are available from the HAMMOZ repository (see https://redmine.hammoz.ethz.ch/projects/hammoz, HAMMOZ consortium, 2019).

*Supplement.* The supplement related to this article is available online at: https://doi.org/10.5194/acp-20-1-2020-supplement.

*Author contributions.* TK and KKup designed the outline of the paper. TK wrote the majority of the paper. TK performed all the climate simulations. VVP and KKup generated the emission scenarios for the climate simulations. TK, TM, HK, AL, JT, and KEJL performed the data analysis for the climate simulations and produced the figures. RVD performed the FASST simulations. VVP, KKup, RVD, and TK performed the TM5-FASST data analysis. All the authors contributed to the writing of the paper.

*Competing interests.* The authors declare that they have no conflict of interest.

*Acknowledgements.* The authors wish to acknowledge the CSC-IT Center for Science, Finland, for computational resources. The ECHAM-HAMMOZ model is developed by a consortium composed of the ETH Zürich, Max Planck Institut für Meteorologie, Forschungszentrum Jülich, University of Oxford, Finnish Meteorological Institute, and Leibniz Institute for Tropospheric Research, and is managed by the Center for Climate Systems Modeling (C2SM) at ETH Zürich.

*Financial support.* This research has been supported by the Academy of Finland, Luonnontieteiden ja Tekniikan Tutkimuksen Toimikunta (grant nos. 286613, 308292, 296644, 272041, and 317373). This article has also received funding from ClimaSlow project "Slowing Down Climate Change: Combining Climate Law and Climate Science to Identify the Best Options to Reduce Emissions of Short-Lived Climate Forcers in Developing Countries" (ERC grant agreement no. 678889) under the EU Horizon 2020 research and innovation programme.

*Review statement.* This paper was edited by Toshihiko Takemura and reviewed by three anonymous referees.

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

**Remarks from the typesetter**

TS1    Please provide a short explanation regarding the requested changes that can be forwarded by us to the editor.

TS2    Please see previous remark.

TS3    Please see previous remark.