# Peer review of "Effects of Black Carbon Mitigation on Arctic Climate"

_Atmospheric Chemistry and Physics, 2019_

## Referee Comment (RC1) · Anonymous Referee #1 · 31 Oct 2019

The manuscript submitted by Kuhn et al. assesses how different black carbon mitigation measures in regions expanding from the eight Arctic Council member states to the entire globe influences the Arctic. They find that 60% of the total effect of global mitigations on BC deposition in the Arctic can be achieved by focusing on local measures. While the direct forcing scales with the total amount of BC emission reduction, the effective radiative forcing, influenced by indirect aerosol effects, is much smaller in magnitude, due to cloud effects that balance the direct forcing. They also include an assessment of Arctic health benefits to aerosol reductions, and find that the amount of premature deaths is strongly reduced if also the observer states in addition to the Arctic Council member states reduce their emissions.

General comments:

[Figure]

- Section 3.1 is a bit confusing – some clarifications in the text will do much to improve this section

- The discussions of how BC and OC influence clouds are solely focused on indirect aerosol effects. Firstly, a short discussion of how BC and OC particles are thought to act as cloud condensation nuclei would improve these discussions. My impression is that while BC particles can act as CCN when duly coated, it is not thought to be as efficient a CCN as OC, which is hygroscopic to begin with. Even so, certain models have been shown to have very strong indirect aerosol effects from BC. A comment of the validity of such model results could be added here. Secondly, numerous studies have shown that BC potentially cause strong rapid adjustments in clouds, but perhaps more importantly due to its influence on the vertical temperature profile and water vapor levels (semidirect effects) than due to the indirect aerosol effects. This should also be discussed in the paper.

Specific comments:

- P1 L19: This important information is difficult to assess in absolute numbers, as we do not have any reference points. Is it possible to rather give the % change in premature deaths?

- P2 L7: Several studies have pointed to the fact that the direct radiative heating effect of BC is in fact offset by rapid adjustments in clouds. Perhaps consider adding some of these references here, and also mention that semidirect effects may be just as or more important than indirect effects in this regard.

- P2 L14: Please add references after "gas industry".

- P6 L21: Please add "2010 – 2030" before "emission reduction", to be entirely clear that the reader is to look at changes along the x axis in Fig. 2 at this point. The statement that OC and BC emissions are mostly unchanged is also not obvious from Fig. 2 – a change of about 1 Tg/yr in BC from 7.2 or so in 2010 to 6.2 or so in 2030

is after all 10-15%, which is definitely a change. SU seems to change by about 100 to about 80 over the same period, which is around 20%, so in other words relatively comparable to the BC change? I realize that I may be misunderstanding here, but if so that just underlines the need for some clarifying comments in the text.

- P6 L22: Please specify in the text that no information on $CO_2$ can be found in Fig. 2.

- P7 L1: See two comments up – this information is excellent, but it would help the reader to provide it further up in the text. Please also give percentage changes for BC and OC emission reductions, for instance for CLE and GLOB, respectively?

- P12 L31: Somewhere in this section, it would be good to see a discussion of how well SU, BC and OC function as CCN in the ECHAM model.

- P14 L8: "the warming effect of SU reductions" → "the direct warming effects of SU reductions", or "the SU warming effects due to aerosol-radiation interactions"

- P14 L16: Please point the reader to where in the vertical profiles we see signs of that upward shift. Also, are we sure that the temperature profile doesn't shift too? If so, there's not necessarily more ice clouds, just liquid clouds higher up. Please comment on this. Here, it would also be nice to mention that, as seen in Fig. 6a moving from CLE to AC – the most local mitigations, which cause strong changes in lower-atmosphere but little changes in upper-atmosphere BC concentrations (Fig. 4), has a very little influence on the direct forcing, presumably due to the low aerosol absorption efficiency at low altitudes.

- P14 L29: This is a good point, concerning the increased aerosol absorption efficiency of BC with height. The literature on this is extensive, so a reference here would be good!

- P14 L30: "the BC mitigation scenarios" is a term not used before – is this equivalent to "SLCF scenarios". If so, make this clear to avoid confusing the reader.

- P14 L30: "Compared to the sRFA values of the BC mitigation scenarios, the sRFA

that is caused by the reductions in SU emissions in the CLE scenario is much smaller in magnitude" Please point the reader to where this small(er) reduction in sRFa can be seen – from Fig. 6a, the sRFa increases from 2010 to 2030, and is reduced in 2050. Which, if any, of these are you referring to?

- P15 L1: "systematically" is perhaps a bit strong – it does not seem that sRFa for CLE and AC follow the BC emissions in the same way as the other scenarios

- P15 L6: Here, on the other hand, I don't agree that there is no visible systematic response in sRFtot to BC/OC reductions: The further away from the Arctic emissions are reduced, the weaker this warming effect in 2030. If I understand correctly, the direct aerosol effect will be included in this tot effect, and as you saw in panel a, this component is more strongly negative moving from AC towards GLOB. Thus, any positive radiative forcing effect is offset the most in GLOB, which causes the weakest positive sRFtot values here.

- P16 L1: "Here the effect of both SU reduction in the CLE scenario and BC and OC reduction in the SLCF scenarios are clearly visible." Could it possibly be also that the small (in relative terms, but perhaps noticeable in absolute terms) reduction in SU between the CLE and the other scenarios is also contributing do this change in CDNC?

- P16 L12: While the CDNC explanation for the increase in sRFtot is plausible, BC have also been shown to have a strong influence on clouds (rapid adjustments / semidirect effects) via the influence on atmospheric heating. Although this element is not included in the analyses, I think the authors should discuss how it could also contribute to explain the findings.

- P16 L13: By "cloud time" in this section – do you mean "cloud lifetime"?

- P16 L28: Here it would be good to mention that this supports earlier findings of BC climate effects – that rapid adjustments in clouds tend to balance out the direct effect.

- P18 L1: here you could add ", which cause lower-level changes in BC, " or something

similar behind "the Arctic Council member states"

Figure 5: Just a simple edit: could you perhaps change the unit of the cloud cover to %? I believe the units on the x axis would be the same, so that you could just change "x10-2" to "(%)".

---

## Referee Comment (RC2) · Anonymous Referee #2 · 6 Dec 2019

**General comment**

The here presented manuscript describes the potential impact of BC-emission reduction on the forcing exerted by aerosol in the Arctic. The topic of the paper is of great interest and represents a nice bridging example between the science and policy communities. Overall the manuscript is well written and clear, however, some editing is still needed. I thus recommend the authors to consider the comments below for a second review round.

**Major comments**

The manuscript tends to be over-inclusive. Some sections are definitely out of context here, especially Section 3.6 is completely out of topic, as also indicated by the absence

of figures in the main text. I suggest removing the above-mentioned section because it does not comply with climatic effects. Something similar might be said for Sections 3.1 and 3.4.1. Global emissions and resulting forcing are definitely of interest, but it is hard to understand what is the goal of such two chapters.

My major concern is, however, the superficial description of the aerosol-physics module (Section 2.2). As a consequence, it becomes almost impossible to judge the reliability of simulations shown in Figure 3, 7 and 9. This is unfortunately amplified by a consistent lack of references along the entire manuscript. Each statement on aerosol-cloud properties or model performances must be explained with citations or results (see in specific comments below). Interestingly, the conclusions are full of references, which is quite unusual.

Specific comments

P2L6-7: please provide a reference.

P2L11-14: these two statements are redundant.

P2L15: sea ice extent.

P2L16-29: in this part of the introduction many protocols, documents and meetings are cited. However, any reference cannot be found. As a note, the whole section suffers from a general lack of peer-reviewed references.

P4L14: replace "sections" with "bins" or "modes". Apply to all manuscript.

P4L13-P5L9: I noticed here a tendency to oversimplification of processes. As examples: in which size mode sits BC or OC? Is OC considered to be completely soluble? The text can be kept simple, but more references should be provided, I assume Kokkola was not the only one using the model.

P5L2: what exactly it is meant with "all aerosol particles are assumed to have the same chemical composition"?

P5L16: would this affect the DRE estimations for the 2030 and 2050 estimations? Could the author provide an estimate for this? I am particularly thinking of this publication doi:10.5194/acp-14-537-2014.

P6L20-23: no black line for SU in figure 2; I thought that $CO_2$ emission was assumed to be the same for all simulations, why $CO_2$ should then decrease? In what way a reduction of SU is unfavorable?

P6L27: I would call this increase as "negligible". Following a previous statement at L21 "..OC and BC emissions are mostly unchanged..."

P7L1-2: Why CLE-SU is compared with OC-SLCF? What do you want to demonstrate?

P7L4-7: provide references and try to articulate a fluent story here.

P7L7-14: As also stated in P11L23, meteorology is very important and large-scale circulation and precipitation control the transport of pollutants to the Arctic. How do the vertical structure of the atmosphere vary in your simulations? Is there any constant feature supporting your upper-lower atmosphere distinction (potential temperature, RH, temperature, etc...)?

P8F3: 1) How do your profiles in 2010 (F3) compare with observations? Are these simulations reasonable? Please elaborate in the text. If such profiles are off, all the following results in the manuscript will be biased. 2) Make use of literature to justify your results, there are plenty of studies on transport to the Arctic. 3) Please change the units from $Kg/m3$ to some more traditional units $ug/m3$.

P10F4: How are the burden calculated in F4? From F3 I would guess that BC burden would be higher in LA than in UP for at least 2010. But it appears to be the contrary. There is no consistency between units used in the text and in F4.

P12L21: "N100 are a common proxy for CCN", never heard of it. Please provide a robust explanation and references.

[Figure]

P12L23-26: along with the paper, there is a tendency to rapidly come to conclusions. I do understand that SU correlates well with N100, but here you do not bring any evidence of NPF dominating the SU concentrations…correlation does not always mean causation. Please explain this, potentially in 3.2.3.

P13F5: N100 and CDNC both in 1/cm-3. Are the 2010 profiles similar to reality (see comment on F3)

P14L7 "A decrease in CDNC means that the cloud droplets are on average larger". This is true if the liquid water content is constant, demonstrate the diameter change. Lower CDNC might mean

P18L26: Could you explain the difference between dry deposition and sedimentation?

P20L19-24: Similar to aerosol vertical profiles…what are your aerosol concentrations in snow for the 2010 year, in the range of Doherty?

---

## Referee Comment (RC3) · Anonymous Referee #3 · 20 Dec 2019

This is an interesting and well-written manuscript presenting a model-based study of the potential impact of reductions in black carbon emissions on the Arctic climate, and also on pollution-related mortality. Of particular note is the focus on the impact of limited reductions by only those countries with a direct interest in the Arctic region, and which therefore have a strong likelihood of actually implementing such changes. I'm pleased to recommend the manuscript for publication in ACP, provided the minor issues noted below can be addressed:

[Figure]

**1 General comments**

1. The introduction section seems heavily slanted towards the policy background to the study – while this is important, this section should include a little more of the scientific background on the climate and health impacts of aerosols and BC in particular, and how this study fits into the previous literature regarding global and regional-scale emission reductions.

2. It's unclear how anthropogenic emissions that aren't directly tied to a country are treated in the scenarios – e.g. shipping emissions that occur over the ocean. Are reductions in these included in the mitigation scenarios , and if so how are they included/excluded in the regional reduction scenarios like AC_ACT?

3. Throughout, the terms "lower atmosphere" (LA) and "upper atmosphere" (UA) are used to refer to the regions below and above 450 hPa. This is confusing, as the term "upper atmosphere" is commonly used to refer to much higher regions above the stratosphere and mesosphere (which are conventionally termed the "middle atmosphere"). I would recommend changing these to "lower troposphere" (LT) and either "upper troposphere" (UT, if contributions from the stratosphere and above are negligible) or "rest of atmosphere" (RA/RoA, otherwise).

**2 Specific comments**

**p.5, lines 24–25** The MACC reanalysis only covers a period of 10 years, but these simulations are run for 30 years. Which year(s) of the reanalysis dataset are used for which years of simulation, or is a derived climatology used rather than the reanalysis directly? For future work, the authors should be aware that this dataset is now superseded by the CAMS reanalysis (Inness et al., 2019; 10.5194/acp-19-3515-2019).

[Figure]

**p.6, lines 8–11** If the meteorology is fixed, then any changes in the source–receptor relationships arising from the rapid adjustments in the atmosphere will not be represented. The authors should consider briefly how significant the impact of this is likely to be.

**p.12, line 24** Why must this be mostly NPF rather than the condensation growth of smaller particles to cross the size threshold?

**p.14, lines 1–2** As well as being "fairly small" this change is also not statistically significant given the stated uncertainty (especially in 2050; it's marginal in 2030).

**p.14, line 16** Is this shift in vertical profile statistically significant or not?

**p.15, line 5** Since the error bars suggest the change in ERF is not statistically significant (unlike that in RF), it's stretching the data to say the changes "lead to warming" here.

**p.23, line 18** "ECHAM" does not include prognostic aerosol; "ECHAM–HAM" or "ECHAM–HAMMOZ" should be used to refer to the aerosol–climate model.

**p.24, lines 13–17** The model data used in the paper should be available to the reader (either from an archive, or at least by contacting the authors) without having to re-run the entire set of simulations.

**Figures 3, 5** Can some indication of the uncertainty on these vertical profile differences be included, as has been done for some of the other types of plot? Otherwise it's unclear whether changes are statistically significant or not.

**Figures 4, 6, 7, 8, 9** The manner in which the error bars represent uncertainty should be briefly stated in the caption.
* * *

---

## Author Comment (AC1) · 27 Feb 2020

**Answers to referees:**
**Effects of Black Carbon Mitigation on Arctic Climate**

We thank the referees for their interest and their time invested in reviewing our manuscript. Thank you also for the constructive comments you made. Please find our answers to your comments in the sections below.

**Referee 1:**

1.  Section 3.1 is a bit confusing – some clarifications in the text will do much to improve this section

    *We re-wrote Section 3.1 to make it easier to understand for the reader.*

2.  The discussions of how BC and OC influence clouds are solely focused on indirect aerosol effects. Firstly, a short discussion of how BC and OC particles are thought to act as cloud condensation nuclei would improve these discussions. My impression is that while BC particles can act as CCN when duly coated, it is not thought to be as efficient a CCN as OC, which is hygroscopic to begin with. Even so, certain models have been shown to have very strong indirect aerosol effects from BC. A comment of the validity of such model results could be added here. Secondly, numerous studies have shown that BC potentially cause strong rapid adjustments in clouds, but perhaps more importantly due to its influence on the vertical temperature profile and water vapor levels (semidirect effects) than due to the indirect aerosol effects. This should also be discussed in the paper.

    *We have included a more detailed description of how SU, BC, and OC affect cloud activation. We have also added discussion on semi-direct effects of BC through changes in vertical temperature profiles, water vapour, and cloud dynamics.*

3.  P1 L19: This important information is difficult to assess in absolute numbers, as we do not have any reference points. Is it possible to rather give the % change in premature deaths?

    *We added the change in % to the abstract and the main text. Note also that there was a typo in the abstract: the amount is 329000, not 339000. We corrected this as well.*

4.  P2 L7: Several studies have pointed to the fact that the direct radiative heating effect of BC is in fact offset by rapid adjustments in clouds. Perhaps consider adding some of these references here, and also mention that semidirect effects may be just as or more important than indirect effects in this regard.

*As mentioned above, we have added discussion on the importance of semi-direct effects of BC aerosol together with appropriate references.*

5. P2 L14: Please add references after "gas industry".

    *We added references as requested.*

6. P6 L21: Please add "2010 – 2030" before "emission reduction", to be entirely clear that the reader is to look at changes along the x axis in Fig. 2 at this point. The statement that OC and BC emissions are mostly unchanged is also not obvious from Fig. 2 – a change of about 1 Tg/yr in BC from 7.2 or so in 2010 to 6.2 or so in 2030 is after all 10-15%, which is definitely a change. SU seems to change by about 100 to about 80 over the same period, which is around 20%, so in other words relatively comparable to the BC change? I realize that I may be misunderstanding here, but if so that just underlines the need for some clarifying comments in the text.

    *You are absolutely right about the size of the changes in BC and OC. We entirely re-wrote Section 3.1 to make it easier to understand. This includes re-structuring the text and adding percentages of the changes in emission strengths for BC, OC and SU.*

7. P6 L22: Please specify in the text that no information on CO2 can be found in Fig. 2.

    *We entirely removed the discussion about $CO_2$ from this Section to avoid confusion.*

8. P7 L1: See two comments up – this information is excellent, but it would help the reader to provide it further up in the text. Please also give percentage changes for BC and OC emission reductions, for instance for CLE and GLOB, respectively?

    *As stated under comment 6, we rewrote Section 3.1 and included the requested numbers.*

9. P12 L31: Somewhere in this section, it would be good to see a discussion of how well SU, BC and OC function as CCN in the ECHAM model.

    *In the cloud activation routine, SU and OC are treated as fully dissolved compounds with hygroscopicity value k of 0.57 and 0.21, respectively. BC is assumed to be completely insoluble and contributes to cloud droplet activation by facilitating condensation of sulfuric acid to the particle phase and affecting the size of activating cloud droplets. We added this description to the Methods section (Section 2.2).*

10. P14 L8: "the warming effect of SU reductions" → "the direct warming effects of SU reductions", or "the SU warming effects due to aerosol-radiation interactions"

    *Thank you for the input, we changed the text accordingly.*

11. P14 L16: Please point the reader to where in the vertical profiles we see signs of that upward shift. Also, are we sure that the temperature profile doesn't shift too? If so, there's not necessarily more ice clouds, just liquid clouds higher up. Please comment on this. Here, it would also be nice to mention that, as seen in Fig. 6a moving from CLE to AC – the most local mitigations, which cause strong changes in lower-atmosphere but little changes in

upper-atmosphere BC concentrations (Fig. 4), has a very little influence on the direct forcing, presumably due to the low aerosol absorption efficiency at low altitudes.

*We added two figures to the supplementary material, Figures S5 and S6. Here the changes in water and ice clouds are visualised separately. The shift in cloud height we were referring to can be seen from the decrease in water cloud occurrence time (Fig. S5, panels c, g, and k) in the lower part of the atmosphere, a similar decrease in ice cloud occurrence time in the lower part of the atmosphere (Fig. S6, panels c, g, and k), and an increase in ice cloud occurrence time in the upper part of the atmosphere. However, these effects are very small and, as referee #3 remarked statistically not significant. We therefore removed this statement.*
*Concerning the statement about Arctic direct forcing due to BC emission changes, assuming here that the referee is referring to Figure 7a, this is a valid point. We pointed this out in the manuscript and added some references.*

12. P14 L29: This is a good point, concerning the increased aerosol absorption efficiency of BC with height. The literature on this is extensive, so a reference here would be good!

    *We added references to previous literature that have pointed out that the absorption efficiency of BC varies with height.*

13. P14 L30: "the BC mitigation scenarios" is a term not used before – is this equivalent to "SLCF scenarios". If so, make this clear to avoid confusing the reader.

    *Thank you for pointing this out – we corrected the text accordingly.*

14. P14 L30: "Compared to the sRFA values of the BC mitigation scenarios, the sRFA that is caused by the reductions in SU emissions in the CLE scenario is much smaller in magnitude" Please point the reader to where this small(er) reduction in sRFa can be seen – from Fig. 6a, the sRFa increases from 2010 to 2030, and is reduced in 2050. Which, if any, of these are you referring to?

    *We added some clarifications to the text and added the corresponding values.*

15. P15 L1: "systematically" is perhaps a bit strong – it does not seem that sRFa for CLE and AC follow the BC emissions in the same way as the other scenarios

    *Agreed. We changed the wording from "systematically" to "fairly well".*

16. P15 L6: Here, on the other hand, I don't agree that there is no visible systematic response in sRFtot to BC/OC reductions: The further away from the Arctic emissions are reduced, the weaker this warming effect in 2030. If I understand correctly, the direct aerosol effect will be included in this tot effect, and as you saw in panel a, this component is more strongly negative moving from AC towards GLOB. Thus, any positive radiative forcing effect is offset the most in GLOB, which causes the weakest positive sRFtot values here.

    *We agree entirely with the referee about the sRF$_A$ values being most negative for the GLOB scenario and hence the total indirect contribution here must also be strongest – we will*

*include this statement in the main text. However, looking at the sRFTOT values (in some sense a short-wave ERF), the five curves are not ordered according to the amount of emitted BC. For instance, the AC curve (blue) is above the CLE curve (black), which would make sense, because AC emits less BC than CLE, but AC is also above AC_ACT (purple), which emits even less BC. Altogether, the differences in sRFTOT between the different scenarios for the same year are not statistically significant. We added this information to the manuscript.*

17. P16 L1: "Here the effect of both SU reduction in the CLE scenario and BC and OC reduction in the SLCF scenarios are clearly visible." Could it possibly be also that the small (in relative terms, but perhaps noticeable in absolute terms) reduction in SU between the CLE and the other scenarios is also contributing do this change in CDNC?

    *This is of course possible, but looking at the vertical SU profiles, we cannot really verify this assumption. We therefore decided to remain with the conclusion that CDNC changes are mainly caused by BC and OC concentration changes.*

18. P16 L12: While the CDNC explanation for the increase in sRFtot is plausible, BC have also been shown to have a strong influence on clouds (rapid adjustments / semidirect effects) via the influence on atmospheric heating. Although this element is not included in the analyses, I think the authors should discuss how it could also contribute to explain the findings.

    *We added a statement concerning the semi-direct effects to the manuscript.*

19. P16 L13: By "cloud time" in this section – do you mean "cloud lifetime"?

    *"Cloud time" here is the accumulated time that a grid box in the model is actually in cloud. Averaging this quantity over a larger region gives a measure of the cloudiness in that region. We explained this better in the manuscript.*

20. P16 L28: Here it would be good to mention that this supports earlier findings of BC climate effects – that rapid adjustments in clouds tend to balance out the direct effect.

    *We added a statement and references as requested.*

21. P18 L1: here you could add ", which cause lower-level changes in BC, " or something similar behind "the Arctic Council member states" Figure 5: Just a simple edit: could you perhaps change the unit of the cloud cover to %? I believe the units on the x axis would be the same, so that you could just change "x10-2" to "(%)".

    *Thank you for the suggestion, we added this to the text. We also changed the units in the figure to % as requested.*

**Referee #2:**

1. The manuscript tends to be over-inclusive. Some sections are definitely out of context here, especially Section 3.6 is completely out of topic, as also indicated by the absence of figures in the main text. I suggest removing the above-mentioned section because it does not comply with climatic effects. Something similar might be said for Sections 3.1 and 3.4.1. Global emissions and resulting forcing are definitely of interest, but it is hard to understand what is the goal of such two chapters.

   *This article is the outcome of a collaboration of climate lawyers, environmentalists, and climate modellers. Therefore, the manuscript has been written in an attempt to trace the possible climate impact of a political instrument (here the Arctic Council) accounting for all the necessary and resulting changes. This includes the mitigation measures that can/should be implemented, the resulting changes in anthropogenic emission strengths, and finally the expected changes in the atmosphere. The latter includes not only climatic effects but also other effects like, for instance, effects on human health (often labelled as co-benefits). This may not seem very relevant to a climate scientist, but for scientists of other fields and for decision makers this is an important aspect to take into account. Just as an example, constructing an emission scenario that only removes BC is not very realistic, because BC and OC are emitted from the same sources and cannot be separated in reality.*
   *The reason we discuss the global emissions is to show how much emissions can be reduced vs how much this affects the Arctic. At the same time, health effects are much more local and to be able to discuss these effects a discussion of the emissions is necessary.*
   *The global radiative forcing is mainly discussed to show the effects of BC and OC vs SU more clearly, because there is so much uncertainty when looking at the Arctic region alone.*

2. My major concern is, however, the superficial description of the aerosol-physics module (Section 2.2). As a consequence, it becomes almost impossible to judge the reliability of simulations shown in Figure 3, 7 and 9. This is unfortunately amplified by a consistent lack of references along the entire manuscript. Each statement on aerosol-cloud properties or model performances must be explained with citations or results (see in specific comments below). Interestingly, the conclusions are full of references, which is quite unusual.

   *We have elaborated the description of the aerosol-physics module SALSA in the revised manuscript and added references to articles where SALSA has been evaluated as follows:*

   *"Here we use SALSA to solve the aerosol microphysics. SALSA represents aerosols by dividing the aerosol size distribution into 10 size sections (or bins), where the aerosol population is further divided into a soluble and an insoluble sub-population. A detailed description of the SALSA size distribution given in Kokkola et al., (2018), elaborating on the size resolution and which aerosol compounds are treated in which size bin. In the same article, an evaluation of ECHAM-SALSA against satellite and ground based remote sensing instruments, in situ observations of aerosol composition and size distribution as well as aircraft measurements of aerosol composition has been performed. In addition, ECHAM-SALSA has been involved in several model experiments within the AEROCOM initiative, where models are compared against aerosol observations and each other (e.g. Burgos et al., 2020; Kristiansen et al., 2016; Kipling et al., 2016; Tsigaridis et al., 2014). Furthermore,*

*ECHAM-SALSA's capability to simulate aerosol-cloud interactions compared to satellite observations has been evaluated in a previous study by Saponaro et al., (2020)."*

*In addition, we have added references throughout the manuscript, where they were lacking.*

3. P2L6-7: please provide a reference.

   *We expanded the description on how BC affects climate in the introduction quite a bit and added references where they were missing.*

4. P2L11-14: these two statements are redundant.

   *The purpose of these two sentences was to explain how different emission sectors contribute to the total BC emissions in different parts of the globe. We re-formulated the sentences to make this more transparent.*

5. P2L15: sea ice extent.

   *corrected*

6. P2L16-29: in this part of the introduction many protocols, documents and meetings are cited. However, any reference cannot be found. As a note, the whole section suffers from a general lack of peer-reviewed references.

   *We added references to this part of the introduction as requested.*

7. P4L14: replace "sections" with "bins" or "modes". Apply to all manuscript.

   *We usually prefer "section" over "bin", but also have used "bin" in the past. "Modes" on the other hand is rather misleading as it usually used in modal aerosol models, which describe their aerosol size distribution in terms of log-normal modes. However, we comply with the referee and change from "section" to "bin".*

8. P4L13-P5L9: I noticed here a tendency to oversimplification of processes. As examples: in which size mode sits BC or OC? Is OC considered to be completely soluble? The text can be kept simple, but more references should be provided, I assume Kokkola was not the only one using the model.

   *We have re-written our description of the aerosol-physics module and added references throughout the text as mentioned above. Details of chemical compounds in different particles sizes in SALSA can be found in the model description and evaluation paper by Kokkola et al., 2018.*

9. P5L2: what exactly it is meant with "all aerosol particles are assumed to have the same chemical composition"?

   *Within one aerosol size bin, all aerosol compounds are assumed to be internally mixed, which means that in each aerosol particle in that size bin the mass fraction of each*

*compound is exactly the same. As aerosol compounds are also referred to as "chemical species", chemical composition makes sense to us.*

10. P5L16: would this affect the DRE estimations for the 2030 and 2050 estimations? Could the author provide an estimate for this? I am particularly thinking of this publication doi:10.5194/acp-14-537-2014.

    *This is a good point. As greenhouse gases heat the atmosphere, one can expect that many atmospheric processes are affected by this, for instance a warmer atmosphere can hold more water vapour, which may have strong cloud effects and thus affect aerosol concentrations. Furthermore, non-anthropogenic aerosol emissions might (e.g. biogenic emissions) may change. These effects are of course not accounted for when the greenhouse gas concentrations are held constant. We commented on this in the manuscript.*

11. P6L20-23: no black line for SU in figure 2; I thought that CO2 emission was assumed to be the same for all simulations, why CO2 should then decrease? In what way a reduction of SU is unfavorable?

    *We entirely re-wrote Section 3.1. As is obvious from the referee's comment, the discussion about CO2 leads only do misunderstandings and we therefore removed it. Just to clarify here: CO2 is one of the main focuses of the ECLIPSEv5a MITIGATION scenario which was used to construct our CLE scenario. However, as we do not consider changes in CO2 in our simulations (in fact, we fix the CO2 concentrations and don't deal with CO2 emissions at all), it was unnecessary to even mention CO2 here.*
    *As SU is generally attributed a cooling effect on climate, its removal would lead to warming. Therefore, when only thinking about climate alone (neglecting e.g. health effects), an SU emission reduction can be seen as unfavourable. The SLCP reduction scenarios have been constructed with this aspect in mind. We tried to explain this better in the text.*

12. P6L27: I would call this increase as "negligible". Following a previous statement at L21"..OC and BC emissions are mostly unchanged. . ."

    *We agree with the referee and changed the text accordingly.*

13. P7L1-2: Why CLE-SU is compared with OC-SLCF? What do you want to demonstrate?

    *Here we merely wanted to point out where the largest changes in emission strength "happen". As SU mainly comes from different sectors than BC and OC, the emission changes can also be seen between different scenarios / years. However, as stated above, we re-wrote Section 3.1 and hopefully made it less confusing.*

14. P7L4-7: provide references and try to articulate a fluent story here.

    *We added references to Winiger et al. (2019) and Sobhani et al. (2018).*

15. P7L7-14: As also stated in P11L23, meteorology is very important and large-scale circulation and precipitation control the transport of pollutants to the Arctic. How do the

vertical structure of the atmosphere vary in your simulations? Is there any constant feature supporting your upper-lower atmosphere distinction (potential temperature, RH, temperature, etc. . .)?

*The main process that controls the vertical distribution of emitted aerosol close to the source is wet deposition. This means that the aerosol concentrations due to local emission more or less continuously decrease until the cloud top is reached. All aerosol that is transported further up is either removed through the much slower processes of turbulent mixing or sedimentation, or it is transported away horizontally. As we are now talking about the stratosphere (above cloud), this horizontal transport is predominantly poleward. This is also how it works in ECHAM-HAMMOZ. In some sense one could say that the modelled average water cloud top defines the boundary between our defined lower and upper atmosphere (which, of course, depends on latitude as well). This can also be seen in Figure 5b (the CDNC values very quickly decrease above the blue dotted line) and the newly added Figure S5c in the supplement, which clearly shows that the cloud time fraction for water clouds is almost zero above that line.*

16. P8F3: 1) How do your profiles in 2010 (F3) compare with observations? Are these simulations reasonable? Please elaborate in the text. If such profiles are off, all the following results in the manuscript will be biased. 2) Make use of literature to justify your results, there are plenty of studies on transport to the Arctic. 3) Please change the units from Kg/m3 to some more traditional units ug/m3.

*We added the following text to the manuscript*

*" In Kokkola et al., 2018, the BC, OC, and SU vertical profiles modelled by ECHAM-SALSA were compared to several aircraft campaigns. There it was found that ECHAM-SALSA tends to overestimate BC concentrations in the source regions and underestimate BC concentrations at high latitudes. Furthermore, we compared the modelled BC vertical profiles to measurement data from the ATom and HIPPO campaigns (not shown), where the model compares quite well with the observations at all latitudes. The OC and SU modelled concentrations agreed in most cases much better with the observations."*

*We changed the units in Figure 3 from kg/m3 to ng/m3.*

17. P10F4: How are the burden calculated in F4? From F3 I would guess that BC burden would be higher in LA than in UP for at least 2010. But it appears to be the contrary. There is no consistency between units used in the text and in F4.

*The burdens are calculated as sum of the total mass of each species in each vertical layer that belongs either to the lower or upper part of the atmosphere as defined in the manuscript. Mathematically this can be written as*
*burden_spec = sum(c_spec(i)\*dz(i)) ,*
*where i is the level index and dz(i) is the level height. The level height in ECHAM-HAMMOZ grows more or less exponentially with altitude, which makes it difficult to compare the total amounts of aerosol contained in different layers by looking at vertical profiles alone.*
*We changed the units for BC in Figure 4 to reflect what was used in the text.*

18. P12L21: "N100 are a common proxy for CCN", never heard of it. Please provide a robust explanation and references.

   *We added several references to the manuscript.*

19. P12L23-26: along with the paper, there is a tendency to rapidly come to conclusions. I do understand that SU correlates well with N100, but here you do not bring any evidence of NPF dominating the SU concentrations. . .correlation does not always mean causation. Please explain this, potentially in 3.2.3.

   *In the model, 97.5% of sulphur is emitted to the gas phase, so correlation between $N_{100}$ and SU trends is a strong indication that the number concentration is very much controlled by how many new particles are formed by nucleation and then grown to $N_{100}$ size through condensation of sulphuric acid to the particles, i.e. new particle formation (NPF) (e.g. Kerminen et al., 2018; Lee et al., 2019). We clarified this in the manuscript.*

20. P13F5: N100 and CDNC both in 1/cm-3. Are the 2010 profiles similar to reality (see comment on F3)

   *We changed the units for $N_{100}$ in Figure 5.*
   *Cloud properties in ECHAM6.3-HAM2.3 (using the aerosol module M7) have been evaluated in Neubauer et al., 2019. ECHAM-SALSA uses the same tuning parameters and the model variables used as tuning constraints do not change much when switching from M7 to SALSA.*
   *Similarly, aerosol number concentration from ECHAM6.3-HAM2.3 (using M7) have been compared to observations in Watson-Parris et al., 2018 and found to agree fairly well. Direct comparison of ECHAM-SALSA $N_{100}$ values with observations has not been performed yet, but are expected to be fairly similar.*

21. P14L7 "A decrease in CDNC means that the cloud droplets are on average larger". This is true if the liquid water content is constant, demonstrate the diameter change. Lower CDNC might mean

   *We have added two figures to the Supplementary information that visualize this increase in cloud droplet size.*

22. P18L26: Could you explain the difference between dry deposition and sedimentation?

   *In this context, sedimentation is the mass flux due to the slow settling of aerosol particles, while dry deposition is the interception by features of the surface (rough surfaces, trees, buildings, etc.) of particles moving close to the surface. While both of these processes are usually combined into one term, in ECHAM-HAMMOZ they are stored in different outputs.*

23. P20L19-24: Similar to aerosol vertical profiles. . .what are your aerosol concentrations in snow for the 2010 year, in the range of Doherty?

   *As stated in the manuscript, ECHAM-SALSA does not track the BC concentrations in snow. If we would try to estimate a value in the way we did for the $RF_{snow}$ values, this would be*

*merely based on the models used in Jiao et al. to simulate these quantities and would therefore not add any extra value to the manuscript.*

**Referee #3:**

1. The introduction section seems heavily slanted towards the policy background to the study – while this is important, this section should include a little more of the scientific background on the climate and health impacts of aerosols and BC in particular, and how this study fits into the previous literature regarding global and regional-scale emission reductions.

    *We extended the introduction to account for these shortcomings.*

2. It's unclear how anthropogenic emissions that aren't directly tied to a country are treated in the scenarios – e.g. shipping emissions that occur over the ocean. Are reductions in these included in the mitigation scenarios, and if so, how are they included/excluded in the regional reduction scenarios like AC_ACT?

    *This is a good point. For all emissions not covered by ECLIPSE we used the ECHAM-HAMMOZ standard inputs for year 2010. This includes, aircraft emissions, biogenic emissions, and wildfire emissions. Emissions from ships are included in ECLIPSEv5a, but have been held constant between the scenarios, because it is not clear how individual countries can affect these emissions. We added this information to the manuscript.*

3. Throughout, the terms "lower atmosphere" (LA) and "upper atmosphere" (UA) are used to refer to the regions below and above 450 hPa. This is confusing, as the term "upper atmosphere" is commonly used to refer to much higher regions above the stratosphere and mesosphere (which are conventionally termed the "middle atmosphere"). I would recommend changing these to "lower troposphere" (LT) and either "upper troposphere" (UT, if contributions from the stratosphere and above are negligible) or "rest of atmosphere" (RA/RoA, otherwise).

    *Thank you for this input. During writing the manuscript we have been struggling to find terms that reasonably well describe our 2-part division of the atmosphere and finally settled on LA and UA. The terms suggested by the referee are probably better (maybe not quite as catchy), so we decided to change our naming to "lower troposphere (LT)" and "rest of the atmosphere (RA)" as suggested.*

4. p.5, lines 24–25 The MACC reanalysis only covers a period of 10 years, but these simulations are run for 30 years. Which year(s) of the reanalysis dataset are used for which years of simulation, or is a derived climatology used rather than the reanalysis directly? For future work, the authors should be aware that this dataset is now superseded by the CAMS reanalysis (Inness et al., 2019; 10.5194/acp19-3515-2019).

    *Like with the greenhouse gas concentrations, we used the same values for all scenarios for all simulation years. The values chosen were the ones for 2010. In our model the O3 and OH concentrations only affect the oxidation of SO2 into sulfuric acid and hence only play a*

*minor role in this study. Nevertheless, we thank the referee for pointing this out and clarified this in the text. Also, recently the standard inputs to ECHAM-HAMMOZ have been changed from MACC to CAMS, and therefore the outdated oxidant fields are no longer in use.*

5.  p.6, lines 8–11 If the meteorology is fixed, then any changes in the source–receptor relationships arising from the rapid adjustments in the atmosphere will not be represented. The authors should consider briefly how significant the impact of this is likely to be.

    *Good point. If meteorology in the model was not fixed, this would most likely affect the aerosol concentrations. However, we would expect the meteorology changes to mostly affect long-distance transport of aerosols. Health effects, on the other hand, are most noticeable inside or close to the source regions and we therefore think that our results would not change significantly. We elaborated on this in the manuscript.*

6.  p.12, line 24 Why must this be mostly NPF rather than the condensation growth of smaller particles to cross the size threshold?

    *Here we consider the process of new particle formation to include both nucleation of new particles and their growth to CCN sizes (see e.g. Kerminen et al., 2018; Lee et al., 2019). We clarified this in the manuscript.*

7.  p.14, lines 1–2 As well as being "fairly small" this change is also not statistically significant given the stated uncertainty (especially in 2050; it's marginal in 2030).

    *We tested statistical significance using the two-sided Mann-Whitney test (e.g. comparing two distributions of 30 yearly mean values). According to that test, the differences mentioned in the text are statistically significant.*

8.  p.14, line 16 Is this shift in vertical profile statistically significant or not?

    *It is not. We changed the wording in the text to make this clear.*

9.  p.15, line 5 Since the error bars suggest the change in ERF is not statistically significant (unlike that in RF), it's stretching the data to say the changes "lead to warming" here.

    *We tested statistical significance using the two-sided Mann-Whitney test (e.g. comparing two distributions of 30 yearly mean values). The change in the current legislation scenario between 2010 and 2030 and again between 2030 and 2050 is statistically significant. The difference between the different scenarios for the same year (e.g. CLE vs AC for 2030) are not statistically significant. We added this information to the manuscript.*

10. p.23, line 18 "ECHAM" does not include prognostic aerosol; "ECHAM–HAM" or "ECHAM–HAMMOZ" should be used to refer to the aerosol–climate model.

    *Agreed. We corrected this in the text.*

11. p.24, lines 13–17 The model data used in the paper should be available to the reader (either from an archive, or at least by contacting the authors) without having to re-run the entire set of simulations.

   *We agree. This has been an oversight from our part and we added the necessary statements to the data availability section.*

12. Figures 3, 5 Can some indication of the uncertainty on these vertical profile differences be included, as has been done for some of the other types of plot? Otherwise it's unclear whether changes are statistically significant or not.

   *We added a grey shading to all vertical profiles for the year 2010, which indicate the interval between the 10$^{th}$ and the 90$^{th}$ percentile of the data.*

13. Figures 4, 6, 7, 8, 9 The manner in which the error bars represent uncertainty should be briefly stated in the caption.

   *Good point. We added a statement to all captions.*

---

## Author Response (AR2)

**Answers to reviewer comments (second revision): Effects of Black Carbon Mitigation on Arctic Climate**

April 1, 2020

We would like thank the referees again for their interest and their time invested in reviewing our manuscript. There was only one additional request:

"The clarification of the statistical significance or otherwise of certain results is welcome. Especially given that the error bars in the plots show simple standard deviation, rather than a corresponding confidence interval, I would recommend stating (briefly) in the text the nature of the Mann-Whitney significance test performed, as included in the authors' response. "

We addressed this by adding a new subsection to the Methods section, Section 2.5, called "uncertainty intervals and statistical significance".

Additionally, we corrected several typos and grammatical inaccuracies. These corrections do not change the contents or conclusions of the article in any way. Please find attached to this response letter a marked-up version of the revised article, where all changes to the text have been highlighted in red.

Additionally there is a small change in two references in the bibliography ("Arctic Council" was displayed as "Council, A.") and one reference (Sobhani et al., 2018) did not appear in the bibliography earlier due to a typo in the main text. These changes in the bibliography are difficult to highlight correctly, because the bibliography is generated automatically with the help of bibtex.

[revised manuscript text omitted]